# Atlantic Multi-decadal Oscillation Modulates the Relationship Between El Niño-Southern Oscillation and Fire Weather in Australia

Guanyu Liu[1], Jing Li[1], Tong Ying[1]

[1] Department of Atmospheric and Oceanic Sciences, School of Physics, Peking University, Beijing, China, 100871

*Correspondence to*: Jing Li (jing-li@pku.edu.cn)

**Abstract** The El Niño-Southern Oscillation (ENSO) is a crucial driver of fire weather in Australia, with the correlation between ENSO and Australian fire weather intensifying over the past two decades. However, the underlying causes for this change have not been thoroughly investigated. In this study, we utilize reanalysis datasets and numerical model simulations
to demonstrate that the Atlantic Multidecadal Oscillation (AMO) could potentially modulate the ENSO-Australian fire weather relationship. The correlation between ENSO and the Australian Fire Weather Index (FWI) increases from 0.17 to 0.70 as the AMO transitions from its negative to positive phase. This strengthening effect can be attributed to atmospheric teleconnection mechanisms. Specifically, the positive AMO phase, characterized by warming in the North and Tropical Atlantic, generates Rossby wave trains, leading to high pressure systems over Australia. Consequently, local temperature and
wind speed increase, while precipitation decreases. This signal, superimposed on ENSO, serves to amplify the ENSO effect on Australian fire weather.

## 1 Introduction

Australia, recognized as the world's primary fire-prone region, is characterized by highly flammable biota and frequently experiences wildfires during the austral spring and summer seasons (Duc et al., 2018; Cai et al., 2009; Dowdy, 2018). The
catastrophic 2019/20 wildfire season exemplifies this vulnerability (Abram et al., 2021; Arriagada et al., 2020). Australian wildfires significantly impact the environment, human health, and ecological systems (Damany-Pearce et al., 2022; Nguyen et al., 2021; Liu et al., 2022). With the intensification of heatwaves and droughts due to global warming, wildfires are anticipated to become more severe, resulting in far-reaching consequences (Clarke and Evans, 2018; Jain et al., 2021).

Fire weather, a key determinant of wildfire occurrence and spread in Australia, is closely related to the El Niño-Southern Oscillation (ENSO), the predominant climate mode governing moisture and temperature variability (Bradstock, 2010; Clarke et al., 2019; Risbey et al., 2009). Typically, El Niño events exacerbate droughts and fire weather in Australia (Chen et al., 2017; M. Mariani et al., 2016). Notably, the ENSO-Australian fire weather relationship has intensified in the 21st century, yet the underlying causes remain elusive (Michela Mariani et al., 2018).

Decadal climate variability, such as the Atlantic Multi-decadal Oscillation (AMO), may also influence fire weather and impose interannual variability. For instance, Atlantic-induced anomalies significantly impact Pacific sea surface temperature (SST) and circulation changes (Chikamoto et al., 2016; Chikamoto et al., 2020; Lv et al., 2022), accounting for nearly 75% of tropical SST changes during the satellite era (Li et al., 2016). The AMO's shift from a negative to a positive phase in the late 1990s may potentially affect ENSO and its teleconnection patterns, including meteorological conditions in Australia associated with fire weather. This study examines the Atlantic's impact on Australian fire weather and its potential contribution to the ENSO-Australian fire weather relationship, elucidating the possible underlying mechanisms. The findings aim to enhance understanding of Australian fire weather variability and improve wildfire modelling and forecasting in the region.

In this study, we systematically examine the influence of the Atlantic Ocean on Australian fire weather conditions and explore its potential role in modulating the ENSO-Australian fire weather relationship. To achieve this, we first scrutinize the ENSO-fire weather connection under distinct AMO phases, identifying an amplification of this relationship during the positive AMO phase. Subsequently, we construct composite maps of various meteorological variables during contrasting ENSO and AMO phases, utilizing an array of multi-source reanalysis datasets. To further substantiate these findings, we employ numerical simulation experiments, providing robust validation of the observed impacts. Lastly, we endeavour to elucidate the underlying mechanisms responsible for these effects through a comprehensive analysis of the simulation results.

## 2 Data and Methods

### 2.1 Fire weather index, meteorological data, and climate variability index

Fire occurrence and spread are determined by the convergence of sufficient fuel, an ignition source, and suitable weather conditions, comprising the fire triangle: heat, fuel, and oxygen (Krawchuk et al., 2009). In regions with high biomass (abundant fuel) such as Australia, fire occurrence over time is primarily modulated by fuel moisture (i.e., weather) and ignitions (lightning and human activities) (Bradstock, 2010). The Fire Weather Index (FWI) is a numerical rating system that estimates fire intensity based on prevailing weather conditions. It has been demonstrated to be a reliable indicator of fire risk due to its consideration of two key factors: fuel availability and the ease of fire spread. Fuel availability is represented by a component related to drought conditions, which affect the presence and amount of combustible materials in an area, affected by the level of dryness in a given area. The ease of spread is a measure of how quickly and extensively a fire can propagate under specific weather conditions normally measured by surface wind speed (Simpson et al., 2014). By incorporating both of these factors, this provides a comprehensive assessment of the potential danger and intensity of fires.

For obtaining this index, it be explicitly calculated using daily meteorological variables including T2M, relative humidity, TP, and WND10. Daily FWI (0.25° x 0.25°, 1981-2019) is obtained from the historical data of fire danger indices from the Copernicus Emergency Management Service (Copernicus Contractor, 2021). This dataset offers a complete historical reconstruction of meteorological conditions favorable for the initiation, spread, and sustainability of fires. The daily FWI is subsequently converted to monthly FWI for further analysis.


Meteorological variables, including surface temperature, precipitation, and wind speed, play a crucial role in determining the FWI (Lawson, 2008). To analyse the response of these variables to remote sea surface temperature (SST) forcing, we primarily utilize monthly meteorological data from the re-gridded and interpolated ERA5 reanalysis datasets (Hersbach, 2019) obtained from the climate data store (CDS) disks. Following Lawson (2008), we select 2m

temperature (T2M), total precipitation (TP), sea level pressure (SLP), and 10m wind speed (WND10) for the period 1981-2019, with a resolution of 0.1°×0.1°. WND10 is calculated using the zonal (U10) and meridional (V10) components of the wind vector to represent the intensity of the 10 m wind. To compare results from various datasets, we also employ the same variables from the NCEP-NCAR Reanalysis 1 datasets with a spatial resolution of 2.5°×2.5°, MERRA-2 datasets with a spatial resolution of 0.625°×0.625° (Global Modeling and Assimilation Office, 2015a;

Global Modeling and Assimilation Office, 2015b; Kalnay et al., 1996), and re-gridded and interpolated ERA5 reanalysis datasets (Hersbach, 2019) with a spatial resolution of 0.1°×0.1° (Hersbach, 2019) for the period 1959-2019. It is noted that the native spatial resolution of the ERA5 reanalysis dataset is 9km on a reduced Gaussian grid. The data used here has been regridded to a regular lat-lon grid of 0.1x0.1 degree by the CDS. All meteorological variables and climate indices undergo linear detrending to minimize the influence of global warming on the analysis. Utilizing these

reanalysis datasets, we generate composite maps of meteorological variables corresponding to distinct phases of the ENSO and AMO. The objective of these composite maps is to elucidate the modulating effect of the AMO on the influence of ENSO in Australia.

Climate variability indices, such as the monthly AMO and Niño 3.4 indices from 1981 to 2019, obtained from the

NCAR climate data guide (Trenberth, 1997; Trenberth & Stepaniak, 2001; Trenberth & Shea, 2006), are employed to represent the variability of AMO and ENSO, respectively. Additionally, we investigate dataset dependence of the primary results by utilizing various SST datasets, including COBE-SST 2 (COBE), the Met Office Hadley Centre's SST (Hadley), and Kaplan Extended SST v2 (Kaplan), NOAA Extended Reconstruction SST V4 (NOAA V4), and NOAA Extended Reconstruction SST V5 (NOAA V5) (Huang et al., 2015; Huang et al., 2017; Kaplan et al., 1998; Liu et al.,

2015; Rayner et al., 2003). The general trends and variability of AMO and the ENSO-Australian FWI relationship under different AMO phases, as indicated by these datasets, exhibit high consistency (Figure S1-S2), further validating the robustness of the results.

We subsequently computed the meteorological variable composite maps of ENSO (AMO), defined as the meteorological variables of Niño 3.4 (AMO) positive phase years minus those in Niño 3.4 (AMO) negative phase years. The positive phase year of Niño 3.4 is determined when the absolute value of the moving average of the Niño 3.4 index in three months exceeds 0.5℃ for at least five months, and vice versa (Trenberth, 1997). Moreover, basic random fluctuations may contribute to observed decadal shifts and should be considered as a potential factor influencing variable ENSO behaviours. Therefore, to obtain a more robust conclusion, we estimated all p-values by accounting for autocorrelation using the method by Storch and Zwiers (2000).

**2.2 Ocean basin experiments**

To investigate the meteorological responses and underlying physical mechanisms of Atlantic influences on Australia, we conducted a series of ocean basin experiments using the Community Earth System Model-Community Atmosphere Model version 4 (CESM-CAM4) (Gent et al., 2011). The North and Tropical Atlantic Sea Surface Temperature (SST) variabilities were incorporated into the model to assess the response of meteorological variables in Australia to these remote forcings. Specifically, the monthly SST variability from 1979 to 2015 was added to the North Atlantic region (25°N-75°N) and the tropical Pacific region (20°N-20°S), with a 10° buffer zone to the north and south of each region. SST forcings in other regions maintained seasonal varying climatological SSTs. The rationale for distinguishing between the Tropical Atlantic and North Atlantic regions stems from the dissimilar sea surface temperature (SST) variability observed in these two areas (not shown). We initially performed the CAM simulation driven by climatological forcing for eight model years. The restart files for each year served as the initial condition for the eight ensemble members. An ensemble simulation comprising eight members was conducted, and the ensemble means were considered the atmospheric response to SST forcing in the target ocean basin. We refer to this experiment as the Ocean Basin Experiments (OBE), which is consistent with previous studies (Liu et al., 2022; Wang et al., 2018).

Given that the peak season for fire weather in Australia primarily occurs during the local spring (September, October, and November; SON) (Earl & Simmonds, 2017), we selected the model responses to SON North Atlantic SST (25°N-65°N, 10°W-60°W) and Tropical Atlantic SST (0-20°N, 10°W-60°W) in the OBE for further analysis. The OBE results were primarily employed to discern the impact of the AMO and to ascertain the underlying teleconnection pattern between the AMO and the ENSO-Australian FWI relationship.

## 2.3 Significance test

The assessment of the differences between ENSO composites with positive AMO (AMO+) and negative AMO (AMO-) was conducted using Student's t-test to ascertain the statistical significance of these differences. This robust analytical method facilitated the evaluation of the potential impacts of AMO phases on the ENSO-Australian fire weather relationship.

The Student's $t$ is calculated as

$$t = \frac{\bar{x}_1 - \bar{x}_2}{\sigma \sqrt{\frac{1}{n_1} + \frac{1}{n_2}}}$$

where $\bar{x}_1$ and $\bar{x}_2$ are sample means, $n_1$ and $n_2$ are sample sizes for different samples, and σ is the pooled standard deviation, which is calculated as

$$\sigma = \sqrt{\frac{n_1 s_1^2 + n_2 s_2^2}{n_1 + n_2 - 2}}$$

where $s_1$ $and$ $s_2$ are standard deviations for different samples. The test statistic under the null hypothesis has Student's t distribution with $n_1 + n_2 - 2$ degrees of freedom.

## 3 Results

### 3.1 Influence of AMO on the relationship between ENSO and Australian FWI

Previous studies have established a strong correlation between Australian fire weather and ENSO (Harris et al., 2014; Mariani et al., 2016), with a significant positive correlation (R≈0.53, p<0.01) observed between Australian FWI and the Niño 3.4 index. El Niño events, characterized by anomalously positive Niño 3.4 indices, typically result in warmer-than-average temperatures and reduced precipitation across most of Australia during the SON season. These meteorological anomalies create favorable conditions for the ignition and spread of wildfires (Keeley et al., 2022; Littell et al., 2016). Additionally, Australia is influenced by a high-pressure center and enhanced northwest winds, which contribute to the expansion of burnt areas (Clements et al., 2008; Koo et al., 2010).

However, the correlation between ENSO and Australian fire weather is not constant, as it changes over time. Mariani et al. (2018) reported that the impact of ENSO on Southern Hemisphere fire weather, including Australia, has intensified since the

beginning of the 21st century. Our analysis supports this finding, with the ENSO-Australian FWI correlation increasing from 0.34 during 1981-1999 to 0.66 during 2000-2019 (Figure S3a). In contrast, the other two major Southern Hemisphere wildfire regions, South Africa and South America, do not exhibit discernible trends (Figure S3 b & c). Mariani et al. (2018) proposed that this correlation shift is likely due to global warming. However, considering the slowdown or pause in the global warming trend between ~2000 and 2015, this hypothesis seems unwarranted, prompting further investigation into alternative causes. Moreover, the strengthened correlation persists even when analyzing detrended time series, further undermining the attribution of this correlation shift to global warming.

Around 2000, the Atlantic Multidecadal Oscillation (AMO), a significant global decadal climate variability, transitioned from its negative to its positive phase. Atlantic climate variability has been demonstrated to have a broad impact on the global climate system, including the Pacific Ocean (Chikamoto et al., 2016), the Indian Ocean (Xue et al., 2018), and even Antarctica (Ren et al., 2022). Indeed, the ENSO-Australian FWI relationship underwent a dramatic change between different AMO phases, with correlation coefficients increasing from 0.17 to 0.70 between negative and positive AMO phases (Figure 1a). This increase is similar to the correlation shift observed before and after 2000 (Figure S1a), albeit with a more pronounced contrast. We further calculated a running correlation between ENSO and Australian FWI and compared its time series with that of the AMO (Figure 1b). It is evident that during the negative AMO phase, the ENSO-Australia FWI correlation is low, mostly below 0.3, while it increases to above 0.6 or even 0.8 during the positive AMO. The timing of the transition between the two time series aligns well, occurring in the late 1990s, although the correlation time series displays a slight lead, primarily due to the smoothing treatment.

**3.2 Impact of North and Tropical Atlantic on Australian FWI**

In an effort to elucidate the potential reinforcement of the ENSO and Australian FWI relationship by the AMO, we conducted a comprehensive examination of the effects on Australian meteorological conditions, placing particular emphasis on the coherent interplay between ENSO and AMO. To achieve this, we analysed reanalysis datasets and OBE simulation results.

As previously noted, positive ENSO, specifically El Niño events, are associated with higher SLP, increased temperature, and reduced precipitation, which collectively contribute to fire generation (Figure 2a-c). With regard to surface wind, the composite field exhibits the same direction (southeast wind) as the climatological surface wind, thereby playing a crucial role in augmenting wind speed and facilitating wildfire expansion (Figure 2c). This relationship is further substantiated by the composite difference maps of temperature, precipitation, and circulation field between El Niño and La Niña events (not shown).

In our investigation, we first juxtapose El Niño-associated meteorological responses in Australia during positive (Figure 2d-f) and negative AMO phases (Figure 2g-i). Our findings reveal that temperatures increases and precipitation decreases are more pronounced during El Niño events coinciding with a positive AMO phase. This intensified response is particularly evident in central and southern Australia, where the predominant vegetation comprises grasslands and shrublands, which are highly susceptible to ignition and wildfire propagation. Furthermore, SLP and WND10 responses are markedly more robust during positive AMO phases compared to negative phases (Figure 2f & i). Specifically, El Niño events in the positive phase of AMO are characterized by elevated SLP and intensified surface winds. The elevated SLP corresponds to descending airflow, consequently exacerbating hot and dry conditions in biomass-rich regions of Australia. Collectively, these observations suggest that AMO may potentially amplify the relationship between ENSO and FWI.

We further conduct a composite analysis of Australian meteorological fields during positive and negative AMO phases (Figure 2j-l), revealing similarities to ENSO patterns (Figure 2a-c). Specifically, positive AMO corresponds to increased temperatures and decreased precipitation, with significant precipitation changes primarily concentrated in the southern region. Given that wildfires are more intense in southern Australia (Hennessy et al., 2005), these AMO-associated meteorological anomalies contribute to heightened fire weather. Additionally, positive AMO corresponds to easterly wind anomalies in eastern Australia. Due to the topography of the Great Dividing Range in eastern Australia, easterly winds adiabatically sink, leading to increased temperatures and reduced humidity in biomass-rich areas (Kriwoken, 1996), thereby creating a high-temperature, low-humidity environment. These similarities suggest that positive AMO may reinforce the ENSO effect on Australian FWI, resulting in more severe and widespread wildfires.

Furthermore, it is indeed crucial to identify the meteorological factors that most significantly contribute to the strengthened ENSO-FWI relationship under the positive AMO phase. As evidenced, T2M plays a dominant role in reinforcing the ENSO-FWI relationship in Western Australia, whereas TP and SLP mainly influence this relationship in Eastern Australia. This distinction can be partly ascribed to the atmospheric circulation patterns illustrated in Figure 2f and i. During the positive phase of the AMO, warm advection from lower latitudes heats the land in Western Australia, resulting in warmer conditions than those observed in the negative phase. On the other hand, in Eastern Australia, the wind predominantly blows from land to sea with limited water vapor content during the positive phase of the AMO. During the negative phase, however, an increased volume of vapor is transported from the southern sea of Australia, leading to enhanced precipitation in Eastern Australia. We also examined the differences in ENSO events between two periods with distinct AMO phases (1981-1999 and 2000-2019, Figure S4) but found no significant differences in their spatial patterns (Figure S4e & f), indicating that ENSO flavour is an unlikely factor driving the shift in the ENSO-Australian FWI relationship.

Reanalysis datasets encompass numerous physical processes; therefore, to isolate the AMO effect, we further investigate the

responses of major meteorological variables (T2M, TP, SLP, U10, V10) to North and Tropical Atlantic in the OBE. We regress the detrended and normalized SON meteorological variables on the detrended and normalized SON Tropical Atlantic and North Atlantic SST in OBE. The regression coefficients, representing the responses of local meteorological variables to remote SST forcings in the corresponding ocean basin, are depicted in Figure 3. Although AMO modulates the ENSO-FWI relationship in both El Niño and La Niña events (Figure S5), El Niño events may induce more severe fire weather in

Australia compared to La Niña events (Figure 2a-c). Consequently, our subsequent discussion primarily focuses on the modulation of Australian fire weather during El Niño conditions.

For the Tropical Atlantic, anomalously high SST corresponds to increased T2M and decreased TP in southern Australia (Figure 3a-b), the primary wildfire region. This anomaly also associates with anomalously easterly and northerly winds in

eastern Australia (Figure 3c). These winds, influenced by topographic factors, result in increased temperature and reduced humidity. Such meteorological anomalies may diminish moisture in combustible plants and heighten surface dryness, creating favourable conditions for wildfire ignition and propagation. Although the responses of these meteorological variables are relatively weaker to North Atlantic forcing (Figure 3d-f), their change directions concur with those under Tropical Atlantic forcing, i.e., positive North Atlantic SST anomalies correspond to increased T2M and decreased TP in

southern Australia. Furthermore, the influence of the tropical Atlantic appears more pronounced and statistically significant, which may be attributable to its closer proximity to Australia. Nevertheless, the impact of the North Atlantic on precipitation remains statistically significant across southern and western Australia, warranting further consideration.

The consistent results in reanalysis (Figure 2) and OBE (Figure 3) suggest that the positive AMO phase, associated with

warm North and Tropical Atlantic SST anomalies, induces warmer and drier weather in Australia, particularly in the southern region. This induction reinforces the positive ENSO signal on Australian fire weather, thereby enhancing the ENSO-fire weather relationship. Moreover, the AMO's modulation effect on Australian climate remains significant in different periods (1981-2019 and 1959-2019) and various reanalysis datasets (ERA5, NCEP-NCAR, and MERRA-2), further corroborating the robustness of our findings (Figure S6-S8).

**3.3 Possible mechanism accounting for this modulation**

To elucidate the physical processes underlying the Atlantic's impact on Australia, we investigate the responses of the 200 hPa geopotential height (GPH) and stream function (SF) in the North and Tropical Atlantic OBE and the mechanisms by

which the North Atlantic and Tropical Atlantic individually influence the Australian FWI. The 200 hPa stream function is a widely employed diagnostic tool for analyzing Rossby wave propagation in previous studies (An et al. 2023, Li et al., 2019).

Following the methodology of Sardeshmukh and Hoskins (1988), we diagnose the dynamics of Rossby waves by examining the barotropic vorticity equation at 200 hPa, specifically focusing on the Rossby wave source (RWS) that quantifies vorticity forcing associated with low-level convergence and upper-level divergence.

In the case of the Tropical Atlantic, thermal forcing in this region drives changes to the zonal Walker circulation. These

alterations may result in upward vertical motion and localized convection over the Atlantic, with corresponding low-level convergence and upper-level divergence subsequently producing an intensification of the local Hadley circulation (Li et al., 2014; Li et al., 2015). This process enhances upper-level convergence at the descending branch of the Hadley cell (Simpkins et al., 2014), leading to an intensification of the local Hadley circulation and the generation of a significant source of Rossby waves that propagate eastward with the climatological mean flow in the Southern Hemisphere (Figure 4e). This Rossby

wave source is evident over the South Atlantic (30°S, 20°W in Figure 4e), and the corresponding Rossby wave will propagate toward Australia, intensifying high pressure in the region (Figure 4a). The regression coefficients of the 200 hPa stream function further corroborate this Rossby wave propagation from the South Atlantic to Australia (Figure 4c). With sea surface temperature (SST) warming in the tropical Atlantic, the response of the stream function in the upper level above Australia corresponds to a high-pressure center with descending airflow in this region (Figure 4c). In summary, anomalous

deep convection in response to increased SST in the Tropical Atlantic drives anomalous divergence of the large-scale flow that extends away from local heating by modulating the Hadley and Walker circulations. This process has been discussed in detail by Simpkins et al. (2014).

Regarding the North Atlantic, warmer Atlantic temperatures heat the air above, forming a local high-pressure center in the

upper troposphere. This signal generates the Rossby wave source over the North Atlantic (Figure 4e), with the corresponding Rossby wave train propagating from west to east, featuring alternating high and low-pressure centers that culminate in a high-pressure anomaly in Australia (Figure 4b). This high pressure corresponds to descending motions over Australia, characterized by drier and hotter air that is unfavorable for cloud and rain formation. It is worth noting that stationary Rossby waves can cross the equator under the influence of meridional background wind, and their direction and tilt structure depend

on the meridional background wind (Li et al., 2015). Furthermore, the responses of the stream function are in strong accordance with those of GPH, with a high-pressure center above Australia. These responses lend further support to the cross-equator propagation under the influence of North Atlantic SST forcing (35°W, 30°N in Figure 4d). The patterns of regression coefficients (Figure 4 a-d) also correspond well to the equatorial windows and wave guides for Rossby wave propagation in the upper troposphere as identified in previous studies (Li et al., 2019). The southward propagation of Rossby

waves originating from the Atlantic is also supported by previous works (Miller et al., 2007; Zhao et al., 2019), which form the basis of the teleconnection between the North Atlantic and Australia. Previous studies also indicate that the Atlantic Multidecadal Oscillation (AMO) can modulate El Niño–Southern Oscillation (ENSO) effects through similar Rossby wave dynamics (Lin and Li, 2012; Nagaraju et al., 2018). The impact of AMO on ENSO itself has been widely discussed in previous studies, encompassing aspects including its influence on ENSO's amplitude, flavor, and predictability. The AMO is known to force changes in the Walker circulation in the tropical Pacific Ocean, affecting ENSO's amplitude (Levine et al., 2017) by impacting the depth of the equatorial thermocline and the positive feedback effect of the thermocline (Geng et al., 2020). For ENSO's flavor, the positive AMO enhances the zonal sea surface temperature gradient in the central Pacific, strengthening zonal advective feedback and favoring extreme and Central Pacific (CP) El Niño development (Gan et al., 2022; Yu et al., 2015). Regarding ENSO predictability, it is modulated by the Atlantic mean state bias and systematic errors in inter-basin interactions (Chikamoto et al., 2020).

In conclusion, SST forcing in the Tropical Atlantic instigates alterations in the Walker circulation, subsequently influencing the Hadley circulation and the generation of Rossby waves. Simultaneously, elevated temperatures in the North Atlantic affect the atmospheric conditions, resulting in the propagation of Rossby waves. These interconnected processes establish the teleconnection between the Atlantic SST and FWI in Australia. It is also noted that the AMO has the potential to modulate the ENSO effects through analogous Rossby wave dynamics, thereby impacting its amplitude, characteristics, and predictability.

## 4 Conclusions

Fire weather in Australia is closely related to the variability of ENSO. The correlation between fire weather in Australia and the variability of ENSO has intensified over the past two decades, yet the underlying cause remains elusive. Through the analysis of reanalysis datasets and ocean basin experiments utilizing a global climate model, our study posits that the AMO modulates the ENSO-Australian fire weather relationship. The correlation coefficient between ENSO and the Australian FWI escalates from 0.17 to 0.70 as the AMO shifts from negative to positive phases. During positive AMO, El Niño conditions correspond to enhanced temperature increases, reduced precipitation, and intensified surface winds conducive to wildfire generation, and vice versa. Physically, a positive AMO, linked to warmer North and Tropical Atlantic sea surface temperatures (SST), generates a local low-pressure center that propagates south-westward via a Rossby wave train. This wave train reaches Australia as a high-pressure anomaly, inducing descending air and promoting warmer, drier meteorological conditions favourable for wildfire generation. These meteorological anomalies amplify the positive ENSO-induced meteorological changes, augmenting the Australian fire weather response to ENSO. Previous research also indicates that positive AMO, characterized by basin-wide Atlantic warming, triggers an Atlantic-Pacific SST seesaw, reinforcing the

Walker circulation over the Pacific (Chafik et al., 2016; McGregor et al., 2014; Wang et al., 2013). This bolstered Walker circulation may further magnify ENSO's effects on Australian FWI.

In contrast to previous studies, this work elucidates the teleconnection between AMO and Australian climate and fire weather on a decadal scale. By investigating the combined influence of ENSO and AMO on fire weather, we have uncovered noteworthy interactions between these climate modes, which can either enhance or suppress fire weather severity during specific phases. The 2019 Australian mega-fire during the austral spring has been ascribed to the El Niño event, positive Southern Annular Mode, and positive Indian Ocean Dipole event of that year (Abram et al., 2021; Nolan et al., 2020; van Oldenborgh et al., 2021). Our study discloses that this period also coincides with elevated Atlantic SST (Figure S9), which may amplify the Australian atmospheric response to the El Niño event, thereby leading to extended dry seasons and intensified heatwaves. Our study highlights the importance of considering both ENSO and AMO variability in predicting fire weather conditions and provides valuable insights for improved understanding and forecasting of fire risk in Australia. It is worth noting that Pacific decadal variability, such as the Pacific Decadal Oscillation (PDO), plays a crucial role in Australia's climate (Power et al., 1999). However, prior research suggests that Pacific variability may be partially induced by Atlantic variability (Li et al., 2016; Ren et al., 2021), underscoring the latter's significance in Earth's climate system. We also examined the modulation effect of the Interdecadal Pacific Oscillation (IPO) or PDO on ENSO and Australian FWI but found it less pronounced than that of the AMO in both observations and simulations (Figures not shown). Admittedly, our analysis is constrained by the FWI data time span availability, and longer time span data may be less reliable due to inadequate observations. Future research will explore the influence of other ocean basins on Australian fire weather and the ENSO-Australian fire weather relationship.

**Data availability**

The history Fire Weather Index (FWI) data is downloaded from fire danger indices historical data in Copernicus Emergency Service (https://doi.org/10.24381/cds.0e89c522). The meteorological reanalysis data is downloaded from ERA5 monthly averaged data in the climate data store (CDS) (https://doi.org/10.24381/cds.adbb2d47). The NCEP-NCAR Reanalysis 1 data is downloaded from Physical Sciences Library (https://psl.noaa.gov/data/gridded/data.ncep.reanalysis.html). The MERRA-2 data is downloaded from MDISC (https://disc.gsfc.nasa.gov/datasets/M2TMNXFLX_5.12.4/summary?keywords=MERRA2_100.tavgM_2d_flx_Nx and https://disc.gsfc.nasa.gov/datasets/M2TMNXSLV_5.12.4/summary?keywords=MERRA2_100.tavgM_2d_slv_Nx). AMO and Niño 3.4 indexes are obtained from the NCAR climate data guide for the AMO index (https://climatedataguide.ucar.edu/climate-data/atlantic-multi-decadal-oscillation-amo) and Niño 3.4 index (https://climatedataguide.ucar.edu/climate-data/nino-sst-indices-nino-12-3-34-4-oni-and-tni), respectively. The sea surface temperature data is obtained from NOAA Extended Reconstructed Sea Surface Temperature (SST) V4

(https://psl.noaa.gov/data/gridded/data.noaa.ersst.v4.html) NOAA Extended Reconstructed Sea Surface Temperature (SST)

V5                (https://psl.noaa.gov/data/gridded/data.noaa.ersst.v5.html)                COBE-SST                2
(https://psl.noaa.gov/data/gridded/data.cobe2.html), HadISST (https://www.metoffice.gov.uk/hadobs/hadisst/), and Kaplan
Extended SST V2 (https://psl.noaa.gov/data/gridded/data.kaplan_sst.html).

## Author contributions

The paper was written by GY and JL and designed by JL. The data analysis was performed by GY and TY. All the co-authors contributed to the interpretation of results and the improvement of this paper.

## Competing interests

The contact author has declared that neither they nor their co-authors have any competing interests.


## Disclaimer

Publisher's note: Copernicus Publications remains neutral with regard to jurisdictional claims in published maps and institutional affiliations.

## Acknowledgement

We thank ECMWF for providing the ERA5 reanalysis data. We also acknowledge the efforts of CAM4 Working Groups and Support Team for developing and maintaining the CAM4 model.

## Financial support

This study is funded by the National Natural Science Foundation of China (NSFC) Grants No. 41975023.

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

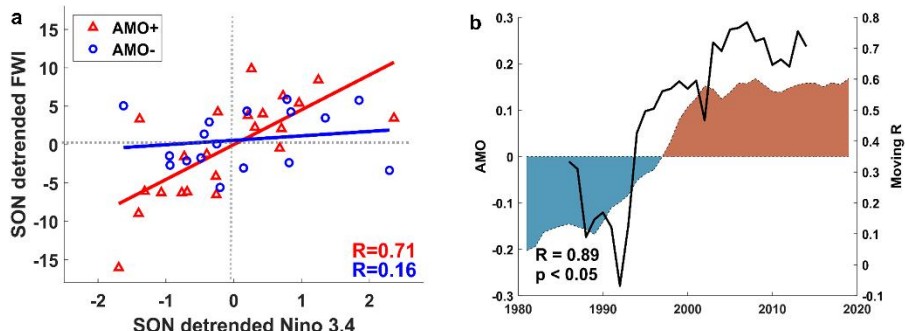

**Figure 1. (a)** Scatter plots for detrended, standardized SON Niño 3.4 index and the corresponding reanalysis mean Australia
FWI from 1981 to 2019. The red upward triangles represent positive AMO indices, while the blue circles represent negative
ones. The lines are linear fit lines. The correlation coefficient (R=0.71) corresponding to AMO+ passed the significance test
of p-value<0.05, while the other one did not. **(b)** The solid black line represents the sliding correlation coefficient between
detrended SON FWI in Australia and detrended SON Niño 3.4 index with a sliding window of 10 years. The shaded area
represents the annual AMO index by detrending and 11-year running mean. Red is positive, and blue is negative. All the
correlation coefficients assume autocorrelation to the time series.

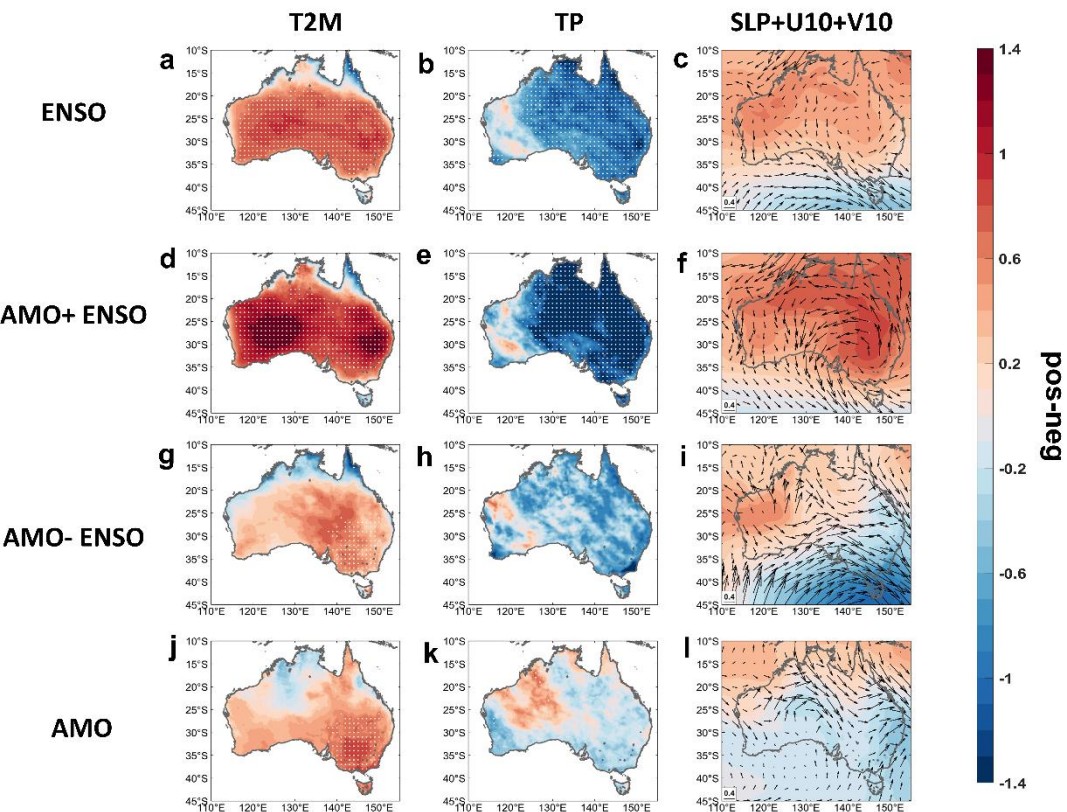

**Figure 2.** The difference maps for the detrended and normalized reanalysis SON **(a, d, g, j)** 2m temperature (T2M), **(b, e, h, k)** Total Precipitation (TP), and **(c, f, i, l)** Sea Level Pressure (SLP)+10m zonal and meridional winds (U10+V10) in conditions with **(a-c)** ENSO composite (El Niño composite minus La Niña composite), **(d-f)** ENSO composite with AMO+, **(g-i)** ENSO composite with AMO-, and **(j-l)** AMO composite from 1981 to 2019. The composite results are calculated using meteorological variables with positive indices minus those with negative ones. The area with white dots passed the significance test of p-value < 0.05 by Student's *t*-test.

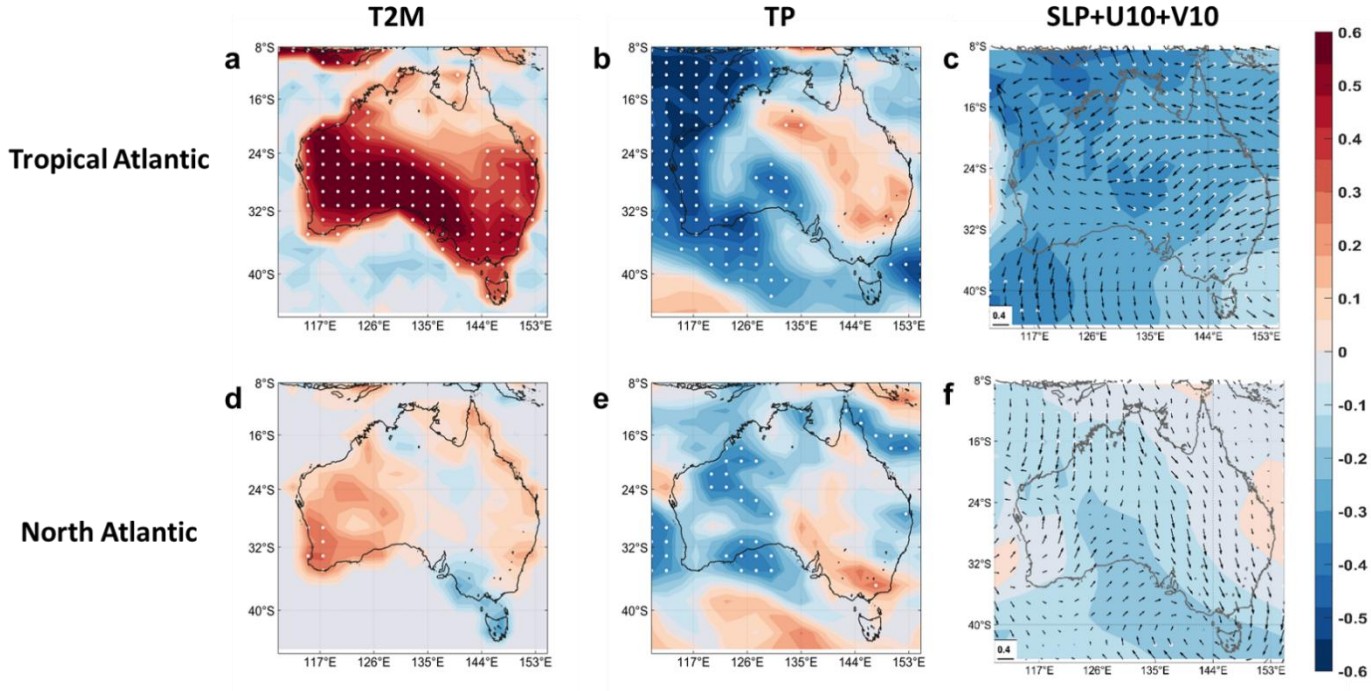

**Figure 3.** Regression coefficients of the ensemble mean detrended and normalized SON **(a, d)** T2M, **(b, e)** TP and **(c, f)** SLP+U10+V10 onto detrended and normalized SON **(a-c)** Tropical Atlantic (10°-60°W, 0-20°N) SST and **(d-f)** North Atlantic (10°-60°W, 25-45°N) SST in the OBE. The area with white dots passed the significance test of $p \leq 0.05$ by Student's *t*-test.

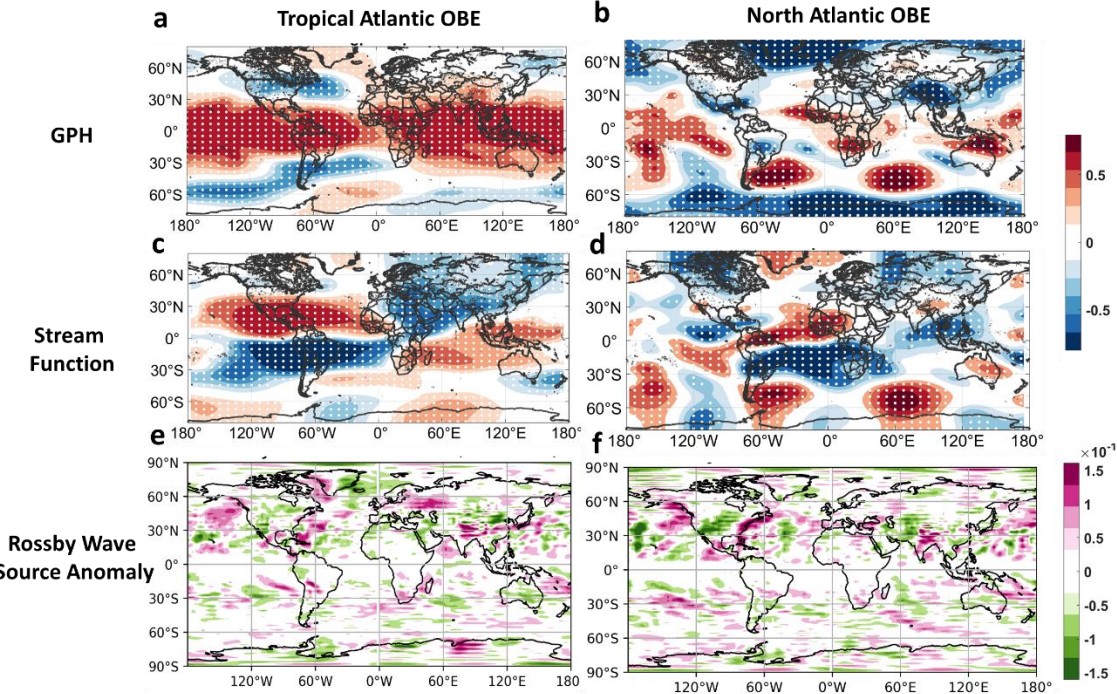

560

**Figure 4. (a-d)** Regression coefficients of detrended and normalized SON 200 hPa **(a-b)** GPH and **(c-d)** stream function onto detrended and normalized SON **(a, c)** Tropical Atlantic (10°-60°W, 0-20°N) and **(b, d)** North Atlantic (10°-60°W, 25-45°N) SST in the OBE. **(e-f)** 200hPa Rossby wave source anomaly in **(e)** Tropical Atlantic OBE, and **(f)** North Atlantic OBE. The area with white dots passed the significance test of p ≤ 0.05 by Student's *t*-test.