# Peer review of "Atlantic Multi-decadal Oscillation Modulates the Relationship Between El Niño-Southern Oscillation and Fire Weather in Australia"

_Atmospheric Chemistry and Physics, 2022_

## Author Comment (AC1)

**Responses to Reviewer 2's comments**

Summary:

This study states that the ENSO-Australian fire relationship can be modulated by the phase changes of AMO. Specially, Atlantic warming may induce warmer temperature and less precipitation, which serves to enhance wildfires when combined with positive ENSO phase. This result is useful in understanding the recent shift in the ENSO-Australian fire relationship and in future wildfire projection in Australia. However, to further improve this manuscript, there are a few issues and questions that need to be addressed。

Reply: We thank this reviewer for the helpful suggestions, which have greatly helped us improve the manuscript. We have studied the comments carefully and made revisions, which we hope will meet the journal standards. Our detailed responses to each of the comments and suggestions are as follows. The referee's original comments are shown in blue. Our replies are shown in black. The corresponding changes in the manuscript are shown in *Italic black*

1. The authors mainly looked at the AMO effect on positive ENSO-Australian fire relationship, i.e., the modulation of Australian fire weather during El Nino conditions. Could they also examine La Nina conditions? Will responses of Australian fire weather also be strengthened during La Nina?

Reply: We appreciate your valuable input. In our analysis, we have indeed examined both El Niño and La Niña conditions during distinct AMO phases, as illustrated in Figure R1. Our findings reveal that the responses of Australian fire weather are not only intensified during El Niño events but also exhibit a similar amplification in La Niña conditions during the positive phase of

AMO.

[Figure]

**Figure R1.** The composite map for the detrended and normalized reanalysis SON FWI in (a) El Niño events when the AMO indexes are positive, (b) La Niña events when AMO indexes are negative, (c) El Niño events when AMO indexes are negative, and (d) La Niña events when AMO indexes are negative. The area with white dots passed the significance test of p-value < 0.05 by Student's *t*-test.

While the AMO indeed modulates the ENSO-FWI relationship in both El Niño and La Niña events, our analysis indicates that El Niño events may result in more severe fire weather in Australia compared to La Niña events. Consequently, this study primarily emphasizes the modulation of Australian fire weather during El Niño conditions.

We added the following discussion concerning the examination on the modulation effect of AMO on La Niña events in Lines 214-217 and cited them here:

*Although AMO modulates the ENSO-FWI relationship in both El Niño and La Niña events (Figure S5), El Niño events may induce more severe fire weather in Australia compared to La Niña events (Figure 2a-c). Consequently, our subsequent discussion primarily focuses on the modulation of Australian fire weather during El Niño conditions.*

2. While it is plausible that AMO modulates the ENSO-Australian fire relationship, another major decadal climate variability, PDO, also shifted its phase around 2000. PDO may exert an even stronger impact on Australian weather. How could the authors exclude the effect of PDO?

Reply: Thank you for highlighting the importance of considering the Interdecadal Pacific Oscillation (PDO) in our analysis. We have indeed investigated the relationship between ENSO and Australia's FWI under various PDO phases using reanalysis data, as well as the Pacific impact on Australia through model simulations. Given the coincident phase transitions of the AMO and PDO in the 1990s, we extended our examination to a longer time series, commencing from 1959, which encompasses a complete PDO and AMO cycle (Figure R2).

Figure R3 presents the ENSO composite maps of Australian meteorological fields under distinct PDO phases, analogous to Figure 2 in the main text. It reveals that the composite maps under varying PDO phases do not exhibit a substantial contrast compared to those under different AMO phases. Although the negative PDO is correlated with marginally stronger temperature (Figure R3 d&g) and precipitation (Figure R3 e&h) alterations, these changes are predominantly statistically insignificant. The SLP and wind fields display minimal contrast (Figure R3 f&i). In comparison, even for the extended time series, the ENSO composites under contrasting AMO phases remain distinct, with a markedly stronger response under the

positive AMO (Figure R3). This outcome suggests that changes in the ENSO-Australia weather relationship under different PDO phases are relatively minor compared to those under the AMO. Nevertheless, as FWI data is only accessible from 1981, our main text will continue to emphasize the period between 1981 and 2019.

[Figure]

**Figure R2.** Time series of monthly AMO and PDO index from 1959 to 2015. The gray dashed line represents zero.

[Figure]

**Figure R3.** The difference maps for the ERA5 detrended and normalized reanalysis SON (a, d, g, j) 2m temperature (T2M), (b, e, h, k) Total Precipitation (TP), and (c, f, I, l) Sea Level Pressure (SLP)+10m zonal and meridional winds (U10+V10) in conditions with (a-c) ENSO composite (El Nino composite minus La Nina composite), (d-f) ENSO composite with PDO+, (g-i) ENSO composite with PDO-, and (j-l) PDO composite from 1959 to 2019. The composite results are calculated using meteorological variables with positive indices minus those with negative ones. The area with white dots passed the significance test of p-value < 0.05 by Student's t-test.

Additionally, we assess the response of Australian meteorological variables to Pacific forcing using ocean basin experiments (OBE). The detrended and normalized SON meteorological variables are regressed on the detrended and normalized SON Tropical and North Pacific SST in OBE. The regression coefficients, representing the responses of local meteorological variables to remote SST forcings in the corresponding ocean basin, are displayed in Figure R4. For the Tropical Pacific SST

anomaly, the responses of T2M, TP, and SLP in Australia do not pass the Student's t-test with a p-value < 0.05. Concerning the North Pacific, the T2M and TP responses are statistically significant in northern Australia (passing the significance test with a p-value < 0.05 by Student's t-test). However, the response magnitudes are relatively small, and the response area is confined to northern Australia. In conclusion, the responses of meteorological fields in Australia to Pacific forcing appear considerably weaker compared to those for Atlantic forcing.

[Figure]

**Figure R4** Regression coefficients of detrended and normalized SON (a, d) T2M, (b, e) TP and (c, f) SLP+U10+V10 onto detrended and normalized SON (a, b) Tropical Pacific (80°-130°W, 0-20°S) SST and North Pacific (170°E-140°W, 25-45°N) SST in the OBE. The area with white dots passed the significance test of p ⩽ 0.05 by Student's *t*-test.

We refrain from concentrating on the PDO in this study, as it is frequently regarded as the interdecadal modulation of high-frequency ENSO variability (Henley et al., 2015). Furthermore, the spatial patterns of surface temperature and precipitation anomalies linked to PDO and ENSO have been reported to exhibit considerable similarities (Deser et al., 2004), rendering the PDO and ENSO potentially indistinguishable in both tropical and extratropical regions (Chen and Wallace, 2015; Zhang et al., 1997). Notably, Atlantic variability may also influence Pacific variability on decadal time

scales (Li et al., 2016), and the PDO phase transition could be partially associated with the AMO.

In summary, given that the PDO's impact on the ENSO-Australia FWI relationship is less pronounced than that of the AMO, and the difficulty in fully distinguishing between the PDO and ENSO, we prioritize examining the AMO in this context. Consequently, we attribute the AMO's phase transition as the primary factor contributing to the heightened ENSO-Australia FWI relationship observed in the 21st century.

We also added the following discussion of the role of PDO in the Lines 313-318 and cited it here:

*It is worth noting that Pacific decadal variability, such as the Pacific Decadal Oscillation (PDO), plays a crucial role in Australia's climate (Power et al., 1999). However, prior research suggests that Pacific variability may be partially induced by Atlantic variability (Li et al., 2016; Ren et al., 2021), underscoring the latter's significance in Earth's climate system. We also examined the modulation effect of the Interdecadal Pacific Oscillation (IPO) or PDO on ENSO and Australian FWI but found it less pronounced than that of the AMO in both observations and simulations (Figures not shown).*

3. The authors only used fire weather to represent fire activities. This may not be exactly equal to the actual fire counts or emissions. It is suggested to validate the correlation between ENSO and Australian fires using other proxies such as burned area, fire counts, etc.

Reply: We appreciate your observation and agree that fire weather may not directly correspond to actual fire counts and burned area. Nevertheless, it is essential to

acknowledge that the majority of data on fire points and burned areas are derived from satellite observations, which have a relatively short temporal range (primarily from 2000 to present). Consequently, employing these datasets to examine the AMO's modulation effect on the ENSO-Australian fire relationship may not be feasible.

In light of this, we have revised our manuscript carefully to primarily concentrate on "fire weather" rather than "fire". This approach allows us to circumvent the limitations posed by the restricted availability of satellite-derived data while maintaining a focus on the broader climatic factors influencing fire-related phenomena in the Australian context.

4. Figure S8 did not give any statistical significance test of SST anomaly, which should be presented for clarity.

Reply: We are grateful for the valuable feedback and have accordingly conducted a statistical significance test for the SST anomaly in Figure S8. For the convenience of the readers, we have also included the corresponding figure here (Figure R5).

[Figure]

**Figure R5.** North Atlantic SST anomaly in **(a)** September, **(b)** October, and **(c)** November 2019. The climatology mean SST is calculated using 1980-2009 SST. The area with white dots passed the significance test of p-value < 0.05 by Student's *t*-test.

**References**

Chen, X. Y., & Wallace, J. M. (2015). ENSO-Like Variability: 1900-2013. *JOURNAL OF CLIMATE*, *28*(24), 9623-9641.

Deser, C., Phillips, A. S., & Hurrell, J. W. (2004). Pacific interdecadal climate variability: Linkages between the tropics and the North Pacific during boreal winter since 1900. *JOURNAL OF CLIMATE*, *17*(16), 3109-3124.

Henley, B. J., Gergis, J., Karoly, D. J., Power, S., Kennedy, J., & Folland, C. K. (2015). A Tripole Index for the Interdecadal Pacific Oscillation. *CLIMATE DYNAMICS*, *45*(11-12), 3077-3090.

Li, X. C., Xie, S. P., Gille, S. T., & Yoo, C. (2016). Atlantic-induced pan-tropical climate change over the past three decades. *Nature Climate Change*, *6*(3), 275-+.

Zhang, Y., Wallace, J. M., & Battisti, D. S. (1997). ENSO-like interdecadal variability: 1900-93. *JOURNAL OF CLIMATE*, *10*(5), 1004-1020.

---

## Author Comment (AC2)

**Responses to Reviewer 5's comments**

Summary:

The authors attempted to explore the underlying mechanism of AMO shift affecting the relationship of ENSO with wildfires in Australia using several reanalysis data and numerical sensitivity simulations. They showed the correlation between ENSO and Australia fire weather index increased from 0.17 to 0.70 when AMO shifted from its negative phase to positive phase. The specific impacting process was that the positive AMO can generate Rossby wave trains and result in the high pressure in Australia and then increased the local temperature and wind speed but decreased precipitation. This paper is well writing and the finding is valuable. I think it can be published after several corrections.

Reply: We thank this reviewer for the helpful suggestions, which have greatly helped us improve the manuscript. We have studied the comments carefully and made revisions, which we hope will meet the journal standards. Our responses to each of the comments and suggestions are as follows. The referee's original comments are shown in blue. Our replies are shown in black. The corresponding changes in the manuscript are shown in *Italic black*

1. This work mainly explored the effect of AMO shift on the wildfires in Australia. It is ok. What about the role of global warming? How to distinguish its role?

Reply: Thanks for your constructive suggestions. Global warming might contribute to this correlation transition, and it is hypothesized to be responsible for this correlation transition by the previous work (Mariani et al., 2018). However, given that the global warming trend slowed down or even paused between ~2000 and 2015, this hypothesis

seems unjustified, motivating us to find other potential causes (decadal climate variability, such as AMO).

In order to assess the influence of global warming on the ENSO-FWI relationship, we constructed a composite map incorporating FWI (fire weather) and TP (total precipitation) variables, as depicted in Figure R1. The responses of FWI and TP are observed to be more pronounced during the positive phase of AMO in comparison to its negative phase. This observation is in strong agreement with the findings presented in Figure 2 of the manuscript, further substantiating that the composite map remains consistent irrespective of whether FWI and meteorological elements are detrended or not. Consequently, it can be inferred that global warming may not play a predominant role in modulating the ENSO-Australian FWI relationship.

Nonetheless, in the original manuscript, all physical quantities have been detrended to account for the potential influence of global warming and ensure that its impact on our analysis is minimized.

[Figure]

**Figure R1.** The composite map for the normalized reanalysis SON (a-d) FWI (fire weather index) and (e-h) TP (total precipitation) in (a, e) El Niño events when the AMO indexes are positive, (b, f) El Niño events when AMO indexes are negative, (c)

La Niña events when AMO indexes are positive, and (d) La Niña events when AMO indexes are negative from 1980 to 2019. The area with white dots passed the significance test of p-value < 0.05 by Student's t-test.

2. The authors highlighted the roles of SST anomalies over the tropical Atlantic and northern Atlantic. The SST effect over tropical Atlantic is ok and the signals are significant. But it is not for north Atlantic, and the correlations between North Atlantic SST and local factors are much weak, in which most of grids have not passed the significant test. So, the main discussions of this paper are suggested to focus on the tropical Atlantic SST anomaly.

Reply: We appreciate your valuable suggestions. The influence of the tropical Atlantic appears more pronounced and statistically significant, which may be attributable to its closer proximity to Australia. Although weaker than the Tropical Atlantic SST, the impact of North Atlantic SST on Australia's precipitation is indeed noteworthy, particularly across southern and western regions (Figure 3e). Moreover, their change directions concur with those under Tropical Atlantic forcing, i.e., positive North Atlantic SST anomalies correspond to increased T2M and decreased TP in southern Australia. Considering the AMO encompasses SST fluctuations in both the tropical and North Atlantic regions, it is imperative to account for the influence of the North Atlantic in our analysis.

We added the following explanation in Lines 223-228 and cited it here:

*Although the responses of these meteorological variables are relatively weaker to North Atlantic forcing (Figure 3d-f), their change directions concur with those under Tropical Atlantic forcing, i.e., positive North Atlantic SST anomalies correspond to increased T2M and decreased TP in southern Australia. Furthermore, the influence of the tropical Atlantic appears more pronounced and statistically significant, which*

*may be attributable to its closer proximity to Australia. Nevertheless, the impact of the North Atlantic on precipitation remains statistically significant across southern and western Australia, warranting further consideration.*

3. To explain the effects of SST anomalies over tropical and north Atlantic on wildfires in Australia, the authors mentioned that the SST anomalies can generate Rossby wave and then affecting the local atmospheric circulations. However, it should be noted that this process involves both hemispheres, and whether the Rossby wave can spread to the southern hemisphere due to trade winds is a matter of debate. The authors can do further analyses to validate this process, such as EP flux, stream function, etc.

Reply: Thanks for your insightful comments and suggestions. We have done further analysis of stream function to further validate this process and modified Figure 4 in the manuscript.

Cross-hemisphere Rossby wave propagation has been substantiated in previous studies, demonstrating the capability of Rossby waves to traverse from the northern to the southern hemisphere (Nagaraju et al., 2018; Zhao et al., 2019). Li et al. (2015) assert that stationary Rossby waves can cross the equator under the influence of meridional background wind, with their direction and tilt structure contingent upon the meridional background wind. To corroborate this process, we have analyzed the regression coefficients of 200 Pa geopotential height and stream function in relation to SST (Figure 4a & b) and Rossby wave source anomaly (Figure 4e & f) in ocean basin experiments for both tropical and North Atlantic regions.

The patterns observed in Figure 4 (a-d) correspond well with the equatorial windows and wave guides for Rossby wave propagation in the upper troposphere, as delineated by Li et al. (2019) in Figure R2. This comprehensive analysis not only supports the

cross-hemisphere propagation of stationary Rossby waves under the influence of meridional background wind, but also contributes to a more in-depth understanding of the underlying processes governing atmospheric dynamics.

[Figure]

**Figure R2.** Schematic diagram for the main CEW ducts. The blue (red) arrows denote the NH-to-SH (SH-to-NH) propagation ducts. The thick arrows indicate high NCEW and thin arrows indicate low NCEW. The equatorial windows and barriers are indicated by a transparent yellow belt around the equator. Darker yellow denotes stronger barrier sections, while transparency denotes windows (The Figure is adapted from Li et al., 2019).

We also extended our discussion in Lines 245-281 and cited it here:

*In the case of the Tropical Atlantic, thermal forcing in this region drives changes to the zonal Walker circulation. These alterations may result in upward vertical motion and localized convection over the Atlantic, with corresponding low-level convergence and upper-level divergence subsequently producing an intensification of the local Hadley circulation (Li et al., 2014; Li et al., 2015). This process enhances upper-level convergence at the descending branch of the Hadley cell (Simpkins et al., 2014), leading to an intensification of the local Hadley circulation and the generation of a significant source of Rossby waves that propagate eastward with the climatological mean flow in the Southern Hemisphere (Figure 4e). This Rossby wave*

source is evident over the South Atlantic (30°S, 20°W in Figure 4e), and the corresponding Rossby wave will propagate toward Australia, intensifying high pressure in the region (Figure 4a). The regression coefficients of the 200 hPa stream function further corroborate this Rossby wave propagation from the South Atlantic to Australia (Figure 4c). With sea surface temperature (SST) warming in the tropical Atlantic, the response of the stream function in the upper level above Australia corresponds to a high-pressure center with descending airflow in this region (Figure 4c). In summary, anomalous deep convection in response to increased SST in the Tropical Atlantic drives anomalous divergence of the large-scale flow that extends away from local heating by modulating the Hadley and Walker circulations. This process has been discussed in detail by Simpkins et al. (2014).

Regarding the North Atlantic, warmer Atlantic temperatures heat the air above, forming a local high-pressure center in the upper troposphere. This signal generates the Rossby wave source over the North Atlantic (Figure 4e), with the corresponding Rossby wave train propagating from west to east, featuring alternating high and low-pressure centers that culminate in a high-pressure anomaly in Australia (Figure 4b). This high pressure corresponds to descending motions over Australia, characterized by drier and hotter air that is unfavorable for cloud and rain formation. It is worth noting that stationary Rossby waves can cross the equator under the influence of meridional background wind, and their direction and tilt structure depend on the meridional background wind (Li et al., 2015). Furthermore, the responses of the stream function are in strong accordance with those of GPH, with a high-pressure center above Australia. These responses lend further support to the cross-equator propagation under the influence of North Atlantic SST forcing (35°W, 30°N in Figure 4d). The patterns of regression coefficients (Figure 4 a-d) also correspond well to the equatorial windows and wave guides for Rossby wave propagation in the upper troposphere as identified in previous studies (Li et al., 2019). The southward propagation of Rossby waves originating from the Atlantic is also supported by

*previous works (Miller et al., 2007; Zhao et al., 2019), which form the basis of the teleconnection between the North Atlantic and Australia. Previous studies also indicate that the Atlantic Multidecadal Oscillation (AMO) can modulate El Niño–Southern Oscillation (ENSO) effects through similar Rossby wave dynamics (Lin and Li, 2012; Nagaraju et al., 2018). The impact of AMO on ENSO itself has been widely discussed in previous studies, including its influence on ENSO's amplitude, flavor, and predictability. The AMO is known to force changes in the Walker circulation in the tropical Pacific Ocean, affecting ENSO's amplitude (Levine et al., 2017) by impacting the depth of the equatorial thermocline and the positive feedback effect of the thermocline (Geng et al., 2020). For ENSO's flavor, the positive AMO enhances the zonal sea surface temperature gradient in the central Pacific, strengthening zonal advective feedback and favoring extreme and Central Pacific (CP) El Niño development (Gan et al., 2022; Yu et al., 2015). Regarding ENSO predictability, it is modulated by the Atlantic mean state bias and systematic errors in inter-basin interactions (Chikamoto et al., 2020).*

4. Line 55-: Is the FWI calculated from these local factors? or it obtained from the other source?

Reply: Thanks for pointing this out. The Fire Weather Index (FWI) employed in this study was obtained from the Copernicus Emergency Service's fire danger indices historical data (https://doi.org/10.24381/cds.0e89c522). This globally recognized index is derived from local meteorological factors, including 2m temperature, relative humidity, and other pertinent variables, to estimate fire danger. The FWI comprises distinct components that encompass the influence of fuel moisture and wind on fire behavior and propagation. An elevated FWI value signifies meteorological conditions that are more conducive to wildfire initiation. A comprehensive description of the FWI calculation methodology is illustrated in Figure R3.

[Figure]

**Figure R3.** Detailed calculation processes of Fire Weather Index (FWI) (The figure is from National Wildfire Coordinating Group, https://www.nwcg.gov/publications/pms437/cffdrs/fire-weather-index-system)

5. Line 75: How to choose the positive and negative phase years of Nino 3.4?

Reply: The positive phase year of the Niño 3.4 index is determined when the absolute value of the three-month moving average surpasses 0.5℃ for a minimum of five months, and the opposite holds true for the negative phase year. This definition is extensively adopted in the National Oceanic and Atmospheric Administration's (NOAA) operational El Niño Southern Oscillation (ENSO) forecasts and numerous prior investigations (Trenberth, 1997). The Niño 3.4 index dataset was acquired from the National Center for Atmospheric Research (NCAR) Climate Data Guide (https://climatedataguide.ucar.edu/climate-data/nino-sst-indices-nino-12-3-34-4-oni-and-tni).

We also added explanation of our choosing processes in Lines 97-98 and cited it here:

*The positive phase year of Niño 3.4 is determined when the absolute value of the*

*moving average of the Niño 3.4 index in three months exceeds 0.5℃ for at least five*

*months, and vice versa (Trenberth, 1997).*

6. Section 2.2: In the OBE, the monthly SST variability is added to the climate mean. However, the early sentence mentioned that the trends were added to the model. Which is right?

Reply: Sorry for the confusion and thanks for pointing this out. The monthly SST variability is added to the climate mean, instead of SST trends. We have revised the sentences in Section 2.2 in Lines 105-108 and cited them here:

*The North and Tropical Atlantic Sea Surface Temperature (SST) variabilities were incorporated into the model to assess the response of meteorological variables in Australia to these remote forcings. Specifically, the monthly SST variability from 1979 to 2015 was added to the North Atlantic region (25°N-75°N) and the tropical Pacific region (20°N-20°S), with a 10° buffer zone to the north and south of each region.*

7. For the OBE, the authors have run eight members. Is there any difference for the initial condition?

Reply: Thanks for pointing this out. These ensemble members have different initial conditions. We first performed CAM simulation forced by climatological forcing for eight model years. The restart files for each year are used as the initial condition of the eight ensemble members.

We also added the explanation of more details of the OBE in Lines 111-113 and cited it here:

*We initially performed the CAM simulation driven by climatological forcing for*

*eight model years. The restart files for each year served as the initial condition for the eight ensemble members.*

8. Line 100: Both the words of "rainfall" and "precipitation" are used in this paper. Strictly, there are some differences between them. Please keep the same expression across this paper.

Reply: You are right and thanks for your helpful suggestions. Precipitation includes liquid and frozen water falling to the Earth's surface. Rainfall only refers to liquid water. We have kept the uniform expression "precipitation" across the paper.

9. For the orders of Figures in this paper, it seemed disorder. For example, Figure 2 are mainly explained first in the main document, rather than Figure 1.

Reply: Thanks for your constructive suggestions. We have modified the main text to ensure that Figure 1 mentioned first.

10. Line 137: This sentence confused me and it not agreed well with the display in Figure 2f and i.

Reply: Sorry for the confusion and thanks for pointing this out. Surface pressure and wind speed increases are both stronger during El Niño when AMO is at its positive phase. We have revised our expression in Lines 182-184 and cited it here:

*Furthermore, SLP and WND10 responses are markedly more robust during positive AMO phases compared to negative phases (Figure 2f & i). Specifically, El Niño events in the positive phase of AMO are characterized by elevated SLP and intensified surface winds.*

11. Line 168: I have not found the results of OBE. To increase readability, please showing the parallel panels of OBE with reanalysis.

Reply: Thanks for your suggestions and sorry for the confusion. We apologize for not showing the parallel panels of OBE with reanalysis. This is mainly because if it is displayed in that way, the number of subplots will be 18, which may reduce the readability of this figure. In turn, we modified the expression citing the corresponding figures to increase its readability in Lines 230-232 and cited it here:

*The consistent results in reanalysis (Figure 2) and OBE (Figure 3) suggest that the positive AMO phase, associated with warm North and Tropical Atlantic SST anomalies, induces warmer and drier weather in Australia, particularly in the southern region.*

**References**

Li, Y. J., Feng, J., Li, J. P., Hu, A. X.: Equatorial windows and barrier for Stationary Rossby waves, J. Clim., 32, 6117-6135, https://doi.org/10.1175/JCLI-D-18-0722.1, 2019

Li, Y. J., Li, J. P., Jin, F. F., Zhao, S.: Interhemispheric propagation of stationary Rossby waves in a horizontally nonuniform background flow. J. Atmos. Sci., 72, 3233-3256, https://doi.org/10.1175/JAS-D-14-0239.1, 2015.

Mariani, M., Holz, A., Veblen, T. T., Williamson, G., Fletcher, M. S., and Bowman, D. M. J. S.: Climate Change Amplifications of Climate-Fire Teleconnections in the Southern Hemisphere, Geophys. Res. Lett., 45(10), 5071-5081, https://doi.org/10.1029/2018GL078294, 2018.

Nagaraju, C., Ashok, K., Balakrishnan Nair, T. M., Guan, Z., and Cai, W.: Potential influence of the Atlantic Multi-decadal Oscillation in modulating the biennial relationship between Indian and Australian summer monsoons, Int. J. Climatol., 38(14), 5220-5230. https://doi.org/10.1002/joc.5722, 2018.

Trenberth, K. E. The definition of El Nino. Bull. Amer. Meteor. Soc, 7$8(12)$, 2771-2778, https://doi.org/10.1175/1520-0477(1997)078<2771:TDOENO>2.0.CO;2, 1997.

Zhao, S., Li, J. P., Li, Y. J., Jin, F. F., and Zheng, J. Y.: Interhemispheric influence of Indo-Pacific convection oscillation on Southern Hemisphere precipitation through southward propagation of Rossby waves, Clim. Dyn., 52(5-6), 3203-3221, https://doi.org/10.1007/s00382-018-4324-y, 2019.

---

## Author Comment (AC3)

**Responses to Reviewer 4's comments**

Summary:

The manuscript "Atlantic Multi-decadal Oscillation Modulates the Relationship Between El Niño-Southern Oscillation and Fire Weather in Australia" studies the relationship between AMO and ENSO-Australian fire weather. This work advances our understanding of the interactions between different climatic phenomena. After carefully revising several issues, I consider this work interesting, relevant, and worth publication in this journal.

Reply: We thank this reviewer for the helpful suggestions, which have greatly helped us improve the manuscript. We have studied the comments carefully and made revisions, which we hope will meet the journal standards. Our responses to each of the comments and suggestions are as follows. The referee's original comments are shown in blue. Our replies are shown in black. The corresponding changes in the manuscript are shown in *Italic black*

1. In general, the manuscript could be enhanced by extending the discussions about the atmospheric teleconnection mechanisms by means of Rossby wave trains. Although the authors do a great job explaining the Rossby wave trains argument, I think the discussion about the Rossby wave source anomalies in Figure 4 (c-d) could be improved to clarify this point to the readers.

Reply: We appreciate your valuable input and have incorporated the suggested changes to enhance our manuscript. In our revised discussion, we have delved deeper into the atmospheric teleconnection mechanism facilitated by Rossby wave trains. This has led to a more comprehensive analysis of the stream function, providing further validation of the underlying processes at play. Consequently, Figure 4 (Figure

R1 here) in the manuscript has been modified to better reflect these findings.

[Figure]

**Figure R1.** (a-d) Regression coefficients of detrended and normalized SON 200 hPa (a-b) GPH and (c-d) stream function onto detrended and normalized SON (a, c) Tropical Atlantic (10°-60°W, 0-20°N) and (b, d) North Atlantic (10°-60°W, 25-45°N) SST in the OBE. (e-f) 200hPa Rossby wave source anomaly in (e) Tropical Atlantic OBE, and (f) North Atlantic OBE. The area with white dots passed the significance test of $p \leq 0.05$ by Student's $t$-test.

Additionally, we have clarified the Rossby wave source anomalies depicted in Figure 4e & f for our readers, ensuring a more coherent understanding of the presented data. The expanded discussion, now encompassing Lines 245-281, not only strengthens our argument but also bolsters the overall quality of the manuscript. We trust that these revisions will prove beneficial in conveying the intricacies of the ENSO-AMO-fire weather relationship.

*In the case of the Tropical Atlantic, thermal forcing in this region drives changes*

to the zonal Walker circulation. These alterations may result in upward vertical motion and localized convection over the Atlantic, with corresponding low-level convergence and upper-level divergence subsequently producing an intensification of the local Hadley circulation (Li et al., 2014; Li et al., 2015). This process enhances upper-level convergence at the descending branch of the Hadley cell (Simpkins et al., 2014), leading to an intensification of the local Hadley circulation and the generation of a significant source of Rossby waves that propagate eastward with the climatological mean flow in the Southern Hemisphere (Figure 4e). This Rossby wave source is evident over the South Atlantic (30 °S, 20 °W in Figure 4e), and the corresponding Rossby wave will propagate toward Australia, intensifying high pressure in the region (Figure 4a). The regression coefficients of the 200 hPa stream function further corroborate this Rossby wave propagation from the South Atlantic to Australia (Figure 4c). With sea surface temperature (SST) warming in the tropical Atlantic, the response of the stream function in the upper level above Australia corresponds to a high-pressure center with descending airflow in this region (Figure 4c). In summary, anomalous deep convection in response to increased SST in the Tropical Atlantic drives anomalous divergence of the large-scale flow that extends away from local heating by modulating the Hadley and Walker circulations. This process has been discussed in detail by Simpkins et al. (2014).

Regarding the North Atlantic, warmer Atlantic temperatures heat the air above, forming a local high-pressure center in the upper troposphere. This signal generates the Rossby wave source over the North Atlantic (Figure 4e), with the corresponding Rossby wave train propagating from west to east, featuring alternating high and low-pressure centers that culminate in a high-pressure anomaly in Australia (Figure 4b). This high pressure corresponds to descending motions over Australia, characterized by drier and hotter air that is unfavorable for cloud and rain formation. It is worth noting that stationary Rossby waves can cross the equator under the influence of meridional background wind, and their direction and tilt structure depend

*on the meridional background wind (Li et al., 2015). Furthermore, the responses of the stream function are in strong accordance with those of GPH, with a high-pressure center above Australia. These responses lend further support to the cross-equator propagation under the influence of North Atlantic SST forcing (35 °W, 30 °N in Figure 4d). The patterns of regression coefficients (Figure 4 a-d) also correspond well to the equatorial windows and wave guides for Rossby wave propagation in the upper troposphere as identified in previous studies (Li et al., 2019). The southward propagation of Rossby waves originating from the Atlantic is also supported by previous works (Miller et al., 2007; Zhao et al., 2019), which form the basis of the teleconnection between the North Atlantic and Australia. Previous studies also indicate that the Atlantic Multidecadal Oscillation (AMO) can modulate El Niño – Southern Oscillation (ENSO) effects through similar Rossby wave dynamics (Lin and Li, 2012; Nagaraju et al., 2018). The impact of AMO on ENSO itself has been widely discussed in previous studies, encompassing aspects including its influence on ENSO's amplitude, flavor, and predictability. The AMO is known to force changes in the Walker circulation in the tropical Pacific Ocean, affecting ENSO's amplitude (Levine et al., 2017) by impacting the depth of the equatorial thermocline and the positive feedback effect of the thermocline (Geng et al., 2020). For ENSO's flavor, the positive AMO enhances the zonal sea surface temperature gradient in the central Pacific, strengthening zonal advective feedback and favoring extreme and Central Pacific (CP) El Niño development (Gan et al., 2022; Yu et al., 2015). Regarding ENSO predictability, it is modulated by the Atlantic mean state bias and systematic errors in inter-basin interactions (Chikamoto et al., 2020).*

2. LN 61-62. I suggest rewriting as: "WND10 is calculated using the zonal (U10) and meridional (V10) components of the wind vector to represent the intensity of the 10 m wind."

Reply: Thanks for your suggestions. We have rewritten this sentence as suggested.

3. LN 62. The authors compared their results with other NCEP-NCAR and MERRA-2 reanalyses "In order to verify the robustness of our results." I am not sure that this comparison ensures the robustness of their results. I would suggest saying instead: "In order to compare LN 63. Please indicate the resolution of the NCEP-NCAR and MERRA-2 reanalysis in the text. Do you use the data in the original grids, or do you regrid to use the same grid as ERA5?

Reply: We appreciate your attention to detail and have incorporated the suggested revisions in our manuscript. The resolution of the NCEP-NCAR and MERRA-2 reanalysis datasets has been explicitly mentioned in the text, ensuring transparency and reproducibility of our methods. Furthermore, we used NCEP-NCAR and MERRA-2 reanalysis in the original grids without regriding.

We modified the expression in Lines 73-79 and cited it here:

*To compare results from various datasets, we also employ the same variables from the NCEP-NCAR Reanalysis 1 datasets with a spatial resolution of 2.5°×2.5°, MERRA-2 datasets with a spatial resolution of 0.625°×0.625° (Global Modeling and Assimilation Office, 2015a; Global Modeling and Assimilation Office, 2015b; Kalnay et al., 1996), and re-gridded and interpolated ERA5 reanalysis datasets with a spatial resolution of 0.1°×0.1° (Hersbach, 2019) for the period 1959-2019. It is noted that the native spatial resolution of the ERA5 reanalysis dataset is 9km on a reduced Gaussian grid. The data used here has been regridded to a regular lat-lon grid of 0.1x0.1 degree by the Climate Data Store (CDS).*

4. Figure S2. Caption: "...and (e,j) Kaplan Extended SST v2 datasets"

Reply: Thanks for your suggestions. We have revised as suggested.

5. LN 115-117. The authors could also mention that the correlations are strengthened even with the analysis of detrended time series. If Global warming alone were the cause, the correlations would disappear.

Reply: Thanks for your helpful suggestion. We have added the explanation as suggested in Lines 153-154 and cited it here:

*Moreover, the strengthened correlation persists even when analyzing detrended time series, further undermining the attribution of this correlation shift to global warming.*

6. LN 140-145. Please improve the explanation of Figure 2.

Reply: Thanks for your helpful suggestions. We have improved the explanation and added more related details of Figure 2 as suggested in Lines 178-186.

We cited them here:

*In our investigation, we first juxtapose El Niño-associated meteorological responses in Australia during positive (Figure 2d-f) and negative AMO phases (Figure 2g-i). Our findings reveal that temperature increases and precipitation decreases are more pronounced during El Niño events coinciding with a positive AMO phase. This intensified response is particularly evident in central and southern Australia, where the predominant vegetation comprises grasslands and shrublands,*

*which are highly susceptible to ignition and wildfire propagation. Furthermore, SLP and WND10 responses are markedly more robust during positive AMO phases compared to negative phases (Figure 2f & i). Specifically, El Niño events in the positive phase of AMO are characterized by elevated SLP and intensified surface winds. The elevated SLP corresponds to descending airflow, consequently exacerbating hot and dry conditions in biomass-rich regions of Australia. Collectively, these observations suggest that AMO may potentially amplify the relationship between ENSO and FWI.*

7. Figure 2 Lowercase i in Caption ".... and (c,f, i, l) Sea Level Pressure ..."

Reply: Thanks for your suggestion. We have revised as suggested.

8. Figure 2. It is not clear how you do the significance test on these figures. I recommend enhancing the manuscript (or in the supplementary material) on how you do your significance test.

Reply: Thanks for your suggestions. We have clarified more details in the significance test in section 2.3. We also cited them here (refer to Lines 123-135):

*The assessment of the differences between ENSO composites with positive AMO (AMO+) and negative AMO (AMO-) was conducted using Student's t-test to ascertain the statistical significance of these differences. This robust analytical method facilitated the evaluation of the potential impacts of AMO phases on the ENSO-Australian fire weather relationship.*

*The Student's t is calculated as*

$$t = \frac{\bar{x}_1 - \bar{x}_2}{\sigma\sqrt{\dfrac{1}{n_1} + \dfrac{1}{n_2}}}$$

where $\bar{x}_1$ and $\bar{x}_2$ are sample means, $n_1$ and $n_2$ are sample sizes for different samples, and $\sigma$ is the pooled standard deviation, which is calculated as

$$\sigma = \sqrt{\frac{n_1 s_1^2 + n_2 s_2^2}{n_1 + n_2 - 2}}$$

where $s_1$ and $s_2$ are standard deviations for different samples. The test statistic under the null hypothesis has Student's t distribution with $n_1 + n_2 - 2$ degrees of freedom.

---

## Author Comment (AC4)

**Responses to Reviewer 1's comments**

Summary:

This manuscript explored the shifting correlations between fire weather indicators and ENSO when AMO was in its positive and negative phases, and has shown interesting results about the combined effects of AMO and ENSO when they are in-phase and out-phase. The authors have tried to interpret the underlying mechanisms teleconnecting Atlantic SST, Australian temperature and precipitation, etc. I am gladly noticing the authors have tested several datasets for the same climate parameters in order to get robust results. This study has provided new information on the impacts of ENSO on fire weather in Australia, the method and analysis are generally solid, the writing is OK, and I suggest this manuscript should be accepted after a proper revision.

Reply: We are grateful to the reviewer for their insightful suggestions, which have significantly contributed to the enhancement of our manuscript. We have meticulously examined the comments and made the necessary revisions, hoping that the updated version meets the journal's standards. Our responses to each of the comments and suggestions are as follows. The referee's original comments are shown in blue. Our replies are shown in black. The corresponding changes in the manuscript are shown in *Italic black*

1.  Statistically, individual AMO+ and ENSO+, and the combination of them, are in favor of a fire-prone climate (high temperature and low precipitation, Fig.2 S5 S6 S7), and this part is OK. But more studies are needed when interpreting the teleconnections: AMO was usually defined as the mean SST of the entire North Atlantic Ocean (including both North and Tropical Atlantic, as you defined in this manuscript). However, it's interesting to notice that the responses to North Atlantic Ocean SST are much weaker than those of the Tropical, and sometimes

even insignificant (Fig.3 4a 4b). Could the authors also show the SSTs time series or patterns for both North and Tropical Atlantic in the supplementary, so the readers could get an idea about at least their differences? I assume the close connection with Tropical Atlantic SST could arise from their closeness in latitudes, so the Rossby wave anomaly arising from a higher Tropical Atlantic SST is easier reaching Australia? which was indicated in Fig.4a? Anyway, more interpretations are needed in this section.

Reply: We appreciate the constructive and valuable suggestions provided. In response, we have presented the Sea Surface Temperature (SST) time series for both the North Atlantic (20°N-70°N) and Tropical Atlantic (0°-20°N) regions for comparison in Figure R1 (also in Figure S). The time series of SSTs were derived similarly to the definition of the Atlantic Multidecadal Oscillation (AMO) index, with detrending and 11-year moving average SST anomalies (Trenberth and Shea, 2006). Generally, the positive and negative phases in the two time series exhibit similar time periods. Both regions experienced negative phases during 1980-2000 and positive phases during 2000-2019. However, their trends in the recent decade differ, with SST decreasing from 2008 in the Tropical Atlantic, while remaining fluctuating in the North Atlantic.

Indeed, the correlations between North Atlantic SST and local factors are weaker than those of the Tropical Atlantic SST, which could be attributed to the closer proximity of the Tropical Atlantic to Australia. Nevertheless, the correlation from the North Atlantic should not be overlooked, particularly as the impact of the North Atlantic on Australia's precipitation is statistically significant across southern and western Australia (Figure 3e). Furthermore, the Atlantic Multidecadal Oscillation itself encompasses SST fluctuations in both the tropical Atlantic and the North Atlantic in the middle latitudes; therefore, the influence of the North Atlantic cannot be disregarded. We have included the following explanation in Lines 226-228 and cited it accordingly:

*Furthermore, the influence of the tropical Atlantic appears more pronounced and statistically significant, which may be attributable to its closer proximity to Australia. Nevertheless, the impact of the North Atlantic on precipitation remains statistically significant across southern and western Australia, warranting further consideration.*

2. Did you remove the global SST warming trend from AMO records? Obviously, most AMO indies are simply the regional means of SST, which combine signals of both global SST warming and multidecadal SST variations. I strongly recommend subtracting the 60°N-60°S mean SST from AMO as Trenberth and Shea (2005) did, or subtracting SSTs of the same latitude from North and Tropical Atlantic SST when you were conducting OBE. Reviewer#2 has already noticed this problem and questioned the role of global warming as well.

Reply: We appreciate the reviewer highlighting this aspect. While global warming might contribute to the observed correlation transition, as hypothesized in previous work (Mariani et al., 2018), the slowdown or even pause of the global warming trend between ~2000 and 2015 renders this hypothesis less plausible, prompting us to explore other potential causes, such as decadal climate variability (e.g., AMO).

Our study focuses on the role of decadal variability rather than global warming. Consequently, we indeed have detrended all physical quantities to minimize the impact of global warming on our analysis.

Following the definition in Trenberth and Shea (2006), we have also removed the global SST warming trend from the AMO records and presented the AMO index from 1870 to 2019 in Figure R1. It is worth noting that the AMO index time series in Figure 1b (1980-2019) is a subset of that in Figure R1 (1870-2019). This may inadvertently lead readers to believe that we did not detrend and remove the influence

of global warming.

[Figure]

**Figure R1** AMO times series from 1870 to 2019

To prevent potential misunderstandings, we have incorporated the following explanation in the Data and Methods section, specifically in Lines 79-80, and cited it accordingly:

*All meteorological variables and climate indices undergo linear detrending to minimize the influence of global warming on the analysis.*

**References**

Mariani, M., Holz, A., Veblen, T. T., Williamson, G., Fletcher, M. S., and Bowman, D. M. J. S.: Climate Change Amplifications of Climate-Fire Teleconnections in the Southern Hemisphere, Geophys. Res. Lett., 45(10), 5071-5081, https://doi.org/10.1029/2018GL078294, 2018.
Trenberth, K. E., and Shea, D. J.: Atlantic hurricanes and natural variability in 2005, Geophys. Res. Lett., 33(12), L12704. https://doi.org/10.1029/2006GL026894, 2006.

---

## Author Comment (AC6)

**Responses to Reviewer 3's comments**

Summary:

Liu et al. investigated the role of Atlantic Multi-decadal Oscillation (AMO) in influencing the relationship between El Niño-Southern Oscillation (ENSO) and Australian fire weather, using both reanalysis data and ocean basin experiments (OBE). Their findings suggest that the positive AMO enhances the ENSO-Australian fire weather relationship, potentially through atmospheric teleconnection mechanisms. These results are important for advancing our understanding of fire-climate relationships. However, the approach and data used in the study may be difficult to follow, as it involves analyses from both reanalysis and OBE. It'll also be helpful to provide additional analysis regarding the contributions of individual meteorological factors, such as increased temperature versus wind speed, to the AMO-strengthened ENSO-Australian fire weather relationship.

Reply: We thank this reviewer for the helpful suggestions, which have greatly helped us improve the manuscript. We have studied the comments carefully and made revisions, which we hope will meet the journal standards. Our responses to each of the comments and suggestions are as follows. The referee's original comments are shown in blue. Our replies are shown in black. The corresponding changes in the manuscript are shown in *Italic black*

**General comments**

1. The authors analyzed both the reanalysis and OBE model results, but when each dataset is used and why they are used are not very clear. It would be helpful to have an overview at the end of the Introduction and/or in Methodology to remind the readers of these.

Reply: We appreciate the insightful recommendations provided by the reviewers. In response, we have incorporated a comprehensive summary at the conclusion of sections 2.1 and 2.2 within the Data and Methods segment. This addition serves to elucidate the rationale behind the utilization of each dataset and delineate the specific circumstances in which they are employed, thereby enhancing the clarity and coherence of our manuscript for the readers (refer to Lines 72-75 and Lines 111-113).

We also cited them here:

*Utilizing these reanalysis datasets, we generate composite maps of meteorological variables corresponding to distinct phases of the ENSO and AMO. The objective of these composite maps is to elucidate the modulating effect of the AMO on the influence of ENSO in Australia.*

*The OBE results were primarily employed to discern the impact of the AMO and to ascertain the underlying teleconnection pattern between the AMO and the ENSO-Australian FWI relationship.*

2. Fig. 1a shows the stronger correlation between ENSO and FWI during positive AMO, and Fig. 2 presents increased temperature, reduced precipitation, stronger high pressure, and larger winds. It'll be interesting to show which meteorological factor contributes most to the strengthened ENSO-FWI relationship under positive AMO.

Reply: Thanks for pointing it out. By comparing Figure 2d-f and g-i, it becomes evident that T2M predominantly contributes to strengthening the ENSO-FWI relationship in Western Australia, while TP and SLP primarily influence this relationship in Eastern Australia. This disparity can be partially attributed to the atmospheric circulation patterns depicted in Figure 2f and i. In Western Australia, during the positive phase of the AMO, warm advection from lower latitudes heats the

land, resulting in warmer conditions compared to the negative phase. Conversely, in Eastern Australia, the wind mainly blows from land to sea with limited water vapor content during the positive phase of the AMO. In the negative phase, however, a greater amount of vapor is transported from the southern sea of Australia, leading to increased precipitation in Eastern Australia.

We also expanded our explanation for Figure 2 in Lines 190-197 and cited it here:

*It is indeed crucial to identify the meteorological factors that most significantly contribute to the strengthened ENSO-FWI relationship under the positive AMO phase. As evidenced, T2M plays a dominant role in reinforcing the ENSO-FWI relationship in Western Australia, whereas TP and SLP mainly influence this relationship in Eastern Australia. This distinction can be partly ascribed to the atmospheric circulation patterns illustrated in Figure 2f and i. During the positive phase of the AMO, warm advection from lower latitudes heats the land in Western Australia, resulting in warmer conditions than those observed in the negative phase. On the other hand, in Eastern Australia, the wind predominantly blows from land to sea with limited water vapor content during the positive phase of the AMO. During the negative phase, however, an increased volume of vapor is transported from the southern sea of Australia, leading to enhanced precipitation in Eastern Australia.*

**Specific comments**

1. line 40: I think the approach is not clear at this point (and after Data and Methods section). Maybe it's better to have an overview of the approaches to help readers understand the plan for investigating the Atlantic impact on Australian fire weather.

Reply: Apologies for any confusion, and thank you for your feedback. In light of your

suggestion, we have incorporated an overview of the methodological approaches at the end of the Introduction section, which will assist readers in comprehending the overall plan for investigating the Atlantic Ocean's impact on Australian fire weather conditions. This overview delineates our examination of the ENSO-fire weather connection under distinct AMO phases, the construction of composite maps, the utilization of numerical simulation experiments, and the comprehensive analysis of underlying mechanisms (refer to Lines 40-47). We believe that this addition will enhance the clarity and coherence of the manuscript, providing readers with a more accessible roadmap for navigating the complexities of the study.

We also cited it here:

*In this study, we systematically examine the influence of the Atlantic Ocean on Australian fire weather conditions and explore its potential role in modulating the ENSO-Australian fire weather relationship. To achieve this, we first scrutinize the ENSO-fire weather connection under distinct AMO phases, identifying an amplification of this relationship during the positive AMO phase. Subsequently, we construct composite maps of various meteorological variables during contrasting ENSO and AMO phases, utilizing an array of multi-source reanalysis datasets. To further substantiate these findings, we employ numerical simulation experiments, providing robust validation of the observed impacts. Lastly, we endeavor to elucidate the underlying mechanisms responsible for these effects through a comprehensive analysis of the simulation results.*

2. line 50: Please specify the meteorological factors considered in the FWI (e.g., wind, temperature, etc.)

Reply: Thanks for pointing this out. We have specified the meteorological factors considered in the calculation of FWI in Lines 61-62 and cited it here:

*For obtaining this index, it be explicitly calculated using daily meteorological variables including T2M, relative humidity, TP, and WND10.*

3. line 52: "...fuel availability (drought conditions)" is confusing. Do you mean drought conditions affect fuel availability? Please explain it in detail.

Reply: Thanks for your helpful suggestion. This sentence indicates that the Fire Weather Index (FWI) incorporates a component related to fuel availability, which is influenced by drought conditions. In other words, the FWI considers the presence and amount of combustible material (fuel) that is available for fires, and this availability is affected by the level of dryness or drought in a given area. We have added the explanation as suggested in Lines 54-60 and cited it here:

*The Fire Weather Index (FWI) is a numerical rating system that estimates fire intensity based on prevailing weather conditions. It has been demonstrated to be a reliable indicator of fire risk due to its consideration of two key factors: fuel availability and the ease of fire spread. Fuel availability is represented by a component related to drought conditions, which affect the presence and amount of combustible materials in an area, affected by the level of dryness in a given area. The ease of spread is a measure of how quickly and extensively a fire can propagate under specific weather conditions normally measured by surface wind speed (Simpson et al., 2014). By incorporating both of these factors, this provides a comprehensive assessment of the potential danger and intensity of fires.*

4. line 63: How do you match and compare products with different spatial resolutions?

Reply: Thanks for pointing it out. No additional processing was applied to the raw reanalysis data. We solely visualized the synthesized meteorological variable fields under different ENSO and AMO phases on a map and conducted a comparison of the resulting outcomes. The composite maps therefore remain their original resolution. The observed consistency among the results from various datasets serves to reinforce the credibility and robustness of our conclusions.

5. line 97: Why do you separate into the two regions? As mentioned by reviewer #1, AMO is usually defined by SST over the North Atlantic. It'll be helpful to elaborate on why you separated the two regions.

Reply: Thanks for your constructive suggestions. The rationale for distinguishing between the Tropical Atlantic and North Atlantic regions stems from the dissimilar sea surface temperature (SST) variability observed in these two areas, as illustrated in Figure R1. Notably, the Tropical Atlantic SST began to exhibit a decreasing trend around 2010, whereas the North Atlantic SST continued to fluctuate without a discernible decline. This distinction enables us to isolate the contributions of the tropical and northern segments of the Atlantic, thereby facilitating a more comprehensive understanding of their respective roles in influencing climate dynamics.

[Figure]

**Figure R1** SST anomaly in Atlantic regions from 1980 to 2019. (a) AMO Index

(0-80°N), (b) Tropical Atlantic SST Index (0°-20°N), and (c) North Atlantic SST

Index (20°N-80°N)

We also added the explanation for the separating into two regions in the manuscript in Lines 109-111 and cited it here:

*The rationale for distinguishing between the Tropical Atlantic and North Atlantic regions stems from the dissimilar sea surface temperature (SST) variability observed in these two areas (not shown).*

6. lines 104-105: Have you analyzed the effects of the ENSO or AMO leading the peak fire season on fire weather? For example, the effect of ENSO 3 or more months prior to peak fire season on fire weather in peak fire season. Chen et al. (2016) showed that NINO4 with a lead time of 10 months has a high correlation with burned areas in Australia.

Reply: We acknowledge the importance of considering the potential influence of ENSO and AMO leading the peak fire season on fire weather conditions in Australia. Nevertheless, it ought to be noted that there may be a reduction in the correlation coefficients when analyzing longer leading times pertaining to the fire weather index (FWI), as shown in Figure R2. This contrast with previous findings by Chen et al. (2016), who assessed the burnt area instead of fire weather, which may require a few months to respond to the weather. Consequently, based on the aforementioned reasons, our study focuses on evaluating the impact of ENSO on the concurrent FWI conditions in Australia.

[Figure]

**Figure R2** The distribution of correlation coefficients between Australian FWI and Nino 3.4 index with different leading time.

7. lines 114-115: How does global warming affect the ENSO-FWI relationship? I know the analyses focus on the effect of natural variability, but it'll be interesting to understand how global warming plays a role in modulating the ENSO-fire weather relationship, along with the contribution from AMO. Several reviewers also mentioned this.

Reply: Thanks for your constructive suggestions. Global warming might contribute to this correlation transition, and it is hypothesized to be responsible for this correlation transition by the previous work (Mariani et al., 2018). However, given that the global warming trend slowed down or even paused between ~2000 and 2015, this hypothesis seems unjustified, motivating us to find other potential causes (decadal climate variability, such as AMO).

In order to assess the influence of global warming on the ENSO-FWI relationship, we constructed a composite map incorporating FWI (fire weather) and TP (total

precipitation) variables, as depicted in Figure R2. The responses of FWI and TP are observed to be more pronounced during the positive phase of AMO in comparison to its negative phase. This observation is in strong agreement with the findings presented in Figure 2 of the manuscript, further substantiating that the composite map remains consistent irrespective of whether FWI and meteorological variables are detrended or not. Consequently, it can be inferred that global warming may not play a predominant role in modulating the ENSO-Australian FWI relationship.

Nonetheless, in the original manuscript, all physical quantities have been detrended to exclude the potential influence of global warming and ensure that its impact on our analysis is minimized.

[Figure]

**Figure R3**. The composite map for the normalized reanalysis SON (a-d) FWI (fire weather index) and (e-h) TP (total precipitation) in (a, e) El Niño events when the AMO indexes are positive, (b, f) El Niño events when AMO indexes are negative, (c) La Niña events when AMO indexes are positive, and (d) La Niña events when AMO indexes are negative from 1980 to 2019. The area with white dots passed the significance test of p-value < 0.05 by Student's t-test.

8. lines 146-151: Figures 2j-l show the composite of AMO, while the following sentences mainly describe positive AMO. It would be helpful to include the

composite analysis for positive AMO only and negative AMO only so that there would be evidence supporting the description of positive AMO.

Reply: We appreciate your valuable input. In our analysis, we have indeed examined both El Niño and La Niña conditions during distinct AMO phases, as illustrated in Figure R3. Our findings reveal that the responses of Australian fire weather are not only intensified during El Niño events but also exhibit a similar amplification in La Niña conditions during the positive phase of AMO.

[Figure]

**Figure R4.** The composite map for the detrended and normalized reanalysis SON FWI in (a) El Niño events when the AMO indexes are positive, (b) La Niña events when AMO indexes are negative, (c) El Niño events when AMO indexes are negative, and (d) La Niña events when AMO indexes are negative. The area with white dots passed the significance test of p-value < 0.05 by Student's t-test.

While the AMO indeed modulates the ENSO-FWI relationship in both El Niño and La Niña events, our analysis indicates that El Niño events may result in more severe fire weather in Australia compared to La Niña events. Consequently, this study primarily emphasizes the modulation of Australian fire weather during El Niño conditions.

We added the following discussion concerning the examination on the modulation effect of AMO on La Niña events in Lines 214-217 and cited them here:

*Although AMO modulates the ENSO-FWI relationship in both El Niño and La Niña events (Figure S5), El Niño events may induce more severe fire weather in Australia compared to La Niña events (Figure 2a-c). Consequently, our subsequent discussion primarily focuses on the modulation of Australian fire weather during El Niño conditions.*

9. lines 227-229: I wonder whether any large fire event is associated with ENSO and positive AMO in history. Or how would the results or conclusions change by replacing fire weather with burned areas or fire emissions? The authors demonstrate the effects of ENSO and AMO on fire weather while lacking actual fire events.

Reply: In recent decades, two significant fire events, the 2009 Black Saturday bushfires, the 2015 Sampson Flat bushfire and the 2019-2020 "Black Summer" bushfires, have been associated with ENSO and positive AMO phases. The 2009 Black Saturday bushfires resulted in 173 fatalities and the destruction of over 2,000 houses (https://www.britannica.com/explore/savingearth/the-australian-black-saturday-bushfires-of-2009). The 2015 Samson Flat bushfire occurred in South Australia, burned

around       12,500       hectares,       and       destroyed       27       homes ([https://www.wikiwand.com/en/2015_Sampson_Flat_bushfires](https://www.wikiwand.com/en/2015_Sampson_Flat_bushfires)).       The       2019-2020 Australian bushfires, which affected multiple states, including New South Wales, Victoria, and South Australia, burnt approximately 19 million hectares and claimed 33 lives ([https://wwf.org.au/what-we-do/australian-bushfires/](https://wwf.org.au/what-we-do/australian-bushfires/)). Both events appear to be closely associated with El Niño occurrences during the positive phase of AMO as illustrated in Figure R4.

[Figure]

**Figure R5** Time series of Niño 3.4 and AMO indexes from 1980 to 2019. The shade corresponds to Niño 3.4 index. The black solid line indicates the AMO index. The gray doted line indicates the threshhold for identifying El Niño events. The arrow indicate large fire events in recent decades corresponding to El Niño occurrences during the positive phase of AMO.

However, due to the limited temporal range of satellite-derived datasets (e.g., fire point and burned area, primarily from 2000 to present), assessing AMO's modulation effect on the ENSO-Australian fire relationship may prove challenging.

In light of these limitations, we have revised our manuscript to primarily focus on "fire weather" rather than "fire" events. This approach allows for a more robust examination of the complex interplay between ENSO, AMO, and fire weather in the context of these devastating events, while acknowledging the constraints imposed by available data.

**Technical comments**

1. lines 81-82: "In addition, simple stochastic variability…..behaviors." This sentence is not clear. Please revise it.

Reply: Thanks for your suggestions and sorry for the confusion. We have revised this sentence and cited it here:

*Moreover, basic random fluctuations may contribute to observed decadal shifts and should be considered as a potential factor influencing variable ENSO behaviours.*

2. line 96: Please specify SON in the main text.

Reply: Thanks for your helpful suggestions. We have specified SON in Lines 116-119 and cited it here:

*Given that the peak season for fire weather in Australia primarily occurs during the local spring (September, October, and November; SON) (Earl & Simmonds, 2017),*

*we selected the model responses to SON North Atlantic SST (25°N-65°N, 10°W-60°W)*
*and Tropical Atlantic SST (0-20°N, 10°W-60°W) in the OBE for further analysis.*

3.  lines 130-131: Broken sentence. Please revise it.

Reply: Thanks for your suggestions. We have revised this sentence in Lines 168-170
and cited it here:

*In an effort to elucidate the potential reinforcement of the ENSO and Australian*
*FWI relationship by the AMO, we conducted a comprehensive examination of the*
*effects on Australian meteorological conditions, placing particular emphasis on the*
*coherent interplay between ENSO and AMO.*

4.  lines 183-185: The sentence is confusing ("...., we examine the responses of the
    500 hPa geopotential height (GPH) responses ...). There are two responses in the
    sentence.

Reply: Thanks for pointing it out. We have deleted one "response" in the sentence
and cited the sentence here (refer to Lines 237-239):

*To elucidate the physical processes underlying the Atlantic's impact on*
*Australia, we investigate the responses of the 200 hPa geopotential height (GPH)*
*and stream function (SF) in the North and Tropical Atlantic OBE and the*
*mechanisms by which the North Atlantic and Tropical Atlantic individually*
*influence the Australian FWI.*

**References**

Yang Chen et al,: Environ. Res. Lett. 11, 045001,
https://doi.org/04500110.1088/1748-9326/11/4/045001, 2016

Mariani, M., Holz, A., Veblen, T. T., Williamson, G., Fletcher, M. S., and Bowman, D. M. J. S.:
Climate Change Amplifications of Climate-Fire Teleconnections in the Southern Hemisphere,
Geophys. Res. Lett., 45(10), 5071-5081, https://doi.org/10.1029/2018GL078294, 2018.

---

## Author Comment (AC7)

**Responses to Reviewer 6's comments**

Summary:

This study examines the effect of Atlantic multidecadal variability on the ENSO-fire weather relationship. A strong modulation is identified using observational data and this is explored further using model simulations.

Overall, the study provides important insights but I think would benefit from some further sensitivity testing to ensure results are well understood and robust.

Reply: We thank this reviewer for the insightful and critical suggestions, which have greatly helped us improve the manuscript. We have studied the comments carefully and made revisions, which we hope will meet the journal standards. Our responses to each of the comments and suggestions are as follows. The referee's original comments are shown in blue. Our replies are shown in black. The corresponding changes in the manuscript are shown in *Italic black*

1. The analysis is limited by data availability back to only 1981 which makes assessing the effects of longer-timescale climate variability, like the AMO, challenging. The authors do an admirable job though (including use of other, longer datasets), but could more explicitly acknowledge the limitation of data length.

Reply: We appreciate your insightful suggestions and have endeavored to evaluate the influence of the AMO using extended datasets, such as the ERA5 dataset spanning from 1959 to 2019. However, we must acknowledge the constraints imposed by data time span availability in our analysis. It is worth noting that longer time span data may be subject to increased unreliability due to suboptimal observation quality. To address this limitation, we have incorporated the discussion in Lines 318-320 and cited it here:

*Admittedly, our analysis is constrained by the FWI data time span availability, and longer time span data may be less reliable due to inadequate observations.*

2. There is surprisingly little discussion of the effect of AMO on ENSO itself. Elaboration on the relationship and mechanisms would be useful (e.g. Levine et al., 2017).

Reply: Thanks for your helpful suggestions. Indeed, the interplay between the AMO and ENSO has been extensively examined in prior research, encompassing aspects such as ENSO's amplitude, flavor, and predictability. Concerning amplitude, the AMO has been demonstrated to induce alterations in the Walker circulation within the tropical Pacific Ocean (Levine et al., 2017), modulating ENSO's intensity by affecting the equatorial thermocline depth and the thermocline's positive feedback effect (Geng et al., 2020). With respect to flavor, the positive AMO phase enhances the zonal sea surface temperature gradient in the central Pacific, thereby strengthening zonal advective feedback and promoting the development of extreme and CP El Niño events (Gan et al., 2022; Yu et al., 2015). Lastly, ENSO predictability is influenced by the Atlantic mean state bias and systematic errors arising from inter-basin interactions (Chikamoto et al., 2020).

We have broadened our discussion on the AMO's impact on ENSO in Lines 274-281.

*The impact of AMO on ENSO itself has been widely discussed in previous studies, encompassing aspects including its influence on ENSO's amplitude, flavor, and predictability. The AMO is known to force changes in the Walker circulation in the tropical Pacific Ocean, affecting ENSO's amplitude (Levine et al., 2017) by impacting the depth of the equatorial thermocline and the positive feedback effect of the thermocline (Geng et al., 2020). For ENSO's flavor, the positive AMO enhances the*

*zonal sea surface temperature gradient in the central Pacific, strengthening zonal advective feedback and favoring extreme and Central Pacific (CP) El Niño development (Gan et al., 2022; Yu et al., 2015). Regarding ENSO predictability, it is modulated by the Atlantic mean state bias and systematic errors in inter-basin interactions (Chikamoto et al., 2020).*

3. The analysis is primarily based on linearly detrended data over 1981-2019. Given the high decadal variability in Australian precipitation, for example, I would be slightly concerned that detrending could accidentally reduce natural variability in the data as well. Sensitivity tests where precipitation is not detrended may be useful.

Reply: We appreciate your thoughtful considerations. Given the potential for detrending to inadvertently diminish natural variability in data, we have analyzed the composite map, which includes Fire Weather Index (FWI) and Total Precipitation (TP), as presented in Figure R1. The responses of FWI and TP exhibit greater intensity during the positive phase of the AMO compared to its negative phase. This finding is in strong agreement with Figure 2 in the manuscript, thereby further validating our conclusions.

Nevertheless, in light of the influence of global warming, all physical quantities in the manuscript have been detrended to minimize the impact of global warming on our analysis.

[Figure]

**Figure R1**. The composite map for the normalized reanalysis SON (a-d) FWI (fire weather index) and (e-h) TP (total precipitation) in (a, e) El Niño events when the AMO indexes are positive, (b, f) El Niño events when AMO indexes are negative, (c) La Niña events when AMO indexes are positive, and (d) La Niña events when AMO indexes are negative from 1980 to 2019. The area with white dots passed the significance test of p-value < 0.05 by Student's *t*-test.

4. Observational datasets are available for some of the variables studied here (e.g. Australian gridded climate dataset (AGCD)). It would be worth comparing ERA-5 against AGCD to ensure ERA-5 is performing adequately.

Reply: Thanks for your helpful suggestions. As suggested, we have employed AFCD for comparing with ERA-5 to ensure that ERA-5 is performing adequately. Moreover, we also compared ERA-5 total precipitation datasets against those from CMAP (CPC Merged Analysis of Precipitation), GPCC (Global Precipitation Climatology Centre), GPCP (Global Precipitation Climatology Project), and PRECL (Precipitation Reconstruction Land) to ensure the reliability of the ERA-5 datasets (Figure R2). In all datasets, the total precipitation responses exhibit greater intensity during the positive phase of the AMO as opposed to its negative phase, which is in strong agreement with Figure 2 in the manuscript. These findings further validate that ERA-5 is performing adequately and substantiate the potential role of the AMO in

modulating the ENSO-fire weather relationship in Australia.

[Figure]

**Figure R2**. The composite map for the normalized reanalysis SON total precipitation (TP) applying (a-d) CMAP (CPC Merged Analysis of Precipitation), (e-h) GPCC (Global Precipitation Climatology Centre), (i-l) GPCP (Global Precipitation Climatology Project), (m-p) PRECL (Precipitation Reconstruction Land), and (q-t) AGCD (Australian Gridded Climate Dataset) in (a, e, i, m, q) El Niño events when the AMO indexes are positive, (b, f, j, n, r) El Niño events when AMO indexes are negative, (c, g, k, o, s) La Niña events when AMO indexes are positive, and (d, h, l, p, t) La Niña events when AMO indexes are negative from 1980 to 2019. The area with white dots passed the significance test of p-value $< 0.05$ by Student's t-test.

5. L22: Suggest changing "climate" to "global"

Reply: Thanks for your suggestions. We have revised as suggested.

6. L32-33: This sentence seems redundant.

Reply: Thanks for your suggestions. We have deleted this sentence to avoid redundancy.

7. L38-39: Better as "shifted from negative to positive phase" I think.

Reply: Thanks for your suggestions. We have revised as suggested.

8. L104-105: Worth adding a qualifier in here such as "typically".

Reply: Thanks for your suggestions. We have revised as suggested.

**References**

Chikamoto, Y., Johnson, Z. F., Wang, S. Y. S., McPhaden, M. J., and Mochizuki, T.: El Nino-Southern Oscillation Evolution Modulated by Atlantic Forcing, J. Geophys. Res. Oceans, 125(8), e2020JC016318, https://doi.org/10.1029/2020JC016318, 2020.

Gan, R., Liu, Q., Huang, G. Hu, K. M. and Li, X. C.: Greenhouse warming and internal variability increase extreme and central Pacific El Niño frequency since 1980. Nat. Commun. 14, 394, https://doi.org/10.1038/s41467-023-36053-7, 2023.

Geng, X., W. Zhang, F. Jin, M. F. Stuecker, and A. F. Z. Levine : Modulation of the Relationship between ENSO and Its Combination Mode by the Atlantic Multidecadal Oscillation. J. Climate, 33, 4679–4695, https://doi.org/10.1175/JCLI-D-19-0740.1, 2020.

Levine, A. F. Z., McPhaden, M. J., and Frierson, D. M. W.: The impact of the AMO on multidecadal ENSO variability, Geophys. Res. Lett., 44, 3877-3886, https://doi.org/10.1002/2017GL072524, 2017.

Yu, J., P. Kao, H. Paek, H. Hsu, C. Hung, M. Lu, and S. An.: Linking Emergence of the Central Pacific El Niño to the Atlantic Multidecadal Oscillation. J. Climate, 28, 651–662, https://doi.org/10.1175/JCLI-D-14-00347.1, 2015.